# Erg251 has complex and pleiotropic effects on sterol composition, azole susceptibility, filamentation, and stress response phenotypes

Xin Zhou[1], Audrey Hilk[1], Norma V. Solis[2], Nivea Pereira De Sa[3], Bode M. Hogan[4], Tessa A. Bierbaum[4], Maurizio Del Poeta[3,5,6], Scott G. Filler[2,7], Laura S. Burrack[4], Anna Selmecki ![ORCID][1] *

1 Department of Microbiology and Immunology, University of Minnesota, Minneapolis, Minnesota, United States of America, 2 Division of Infectious Diseases, Lundquist Institute for Biomedical Innovation at Harbor UCLA Medical Center, Torrance, California, United States of America, 3 Department of Microbiology and Immunology, Stony Brook University, Stony Brook, New York, United States of America, 4 Gustavus Adolphus College, Department of Biology, Saint Peter, Minnesota, USA, 5 Division of Infectious Diseases, School of Medicine, Stony Brook University, Stony Brook, New York, United States of America, 6 Veterans Administration Medical Center, Northport, New York, United States of America, 7 David Geffen School of Medicine at UCLA, Los Angeles, California, United States of America

* selmecki@umn.edu

**Data Availability Statement:** All whole genome sequences and RNA sequences are available in the NCBI Sequence Read Archive repositories

## Abstract

Ergosterol is essential for fungal cell membrane integrity and growth, and numerous antifungal drugs target ergosterol. Inactivation or modification of ergosterol biosynthetic genes can lead to changes in antifungal drug susceptibility, filamentation and stress response. Here, we found that the ergosterol biosynthesis gene *ERG251* is a hotspot for point mutations during adaptation to antifungal drug stress within two distinct genetic backgrounds of *Candida albicans*. Heterozygous point mutations led to single allele dysfunction of *ERG251* and resulted in azole tolerance in both genetic backgrounds. This is the first known example of point mutations causing azole tolerance in *C. albicans*. Importantly, single allele dysfunction of *ERG251* in combination with recurrent chromosome aneuploidies resulted in *bona fide* azole resistance. Homozygous deletions of *ERG251* caused increased fitness in low concentrations of fluconazole and decreased fitness in rich medium, especially at low initial cell density. Homozygous deletions of *ERG251* resulted in accumulation of ergosterol intermediates consistent with the fitness defect in rich medium. Dysfunction of *ERG251*, together with FLC exposure, resulted in decreased accumulation of the toxic sterol (14-α-methylergosta-8,24(28)-dien-3β,6α-diol) and increased accumulation of non-toxic alternative sterols. The altered sterol composition of the *ERG251* mutants had pleiotropic effects on transcription, filamentation, and stress responses including cell membrane, osmotic and oxidative stress. Interestingly, while dysfunction of *ERG251* resulted in azole tolerance, it also led to transcriptional upregulation of *ZRT2*, a membrane-bound Zinc transporter, in the presence of FLC, and overexpression of *ZRT2* is sufficient to increase azole tolerance in wild-type *C. albicans*. Finally, in a murine model of systemic infection, homozygous deletion of *ERG251* resulted in decreased virulence while the heterozygous deletion mutants maintain their

BioProject accession numbers PRJNA1068093 and PRJNA1068582.

**Funding:** Funding for this work was supported in part by the National Institute of Allergy and Infectious Diseases (R01 AI143689) and Burroughs Wellcome Fund Investigator in the Pathogenesis of Infectious Diseases Award (#1020388) to AS, an National Institute of Allergy and Infectious Diseases grant AI125770 and the Research Career Scientist Award (IK6 BX005386) from the U.S. Department of Veterans Affairs to MDP, the Swanson-Holcomb Undergraduate Research Fund at Gustavus Adolphus College to TAB and LSB, and the First Year Research Experience Award at Gustavus Adolphus College to BMH. The funders had no role in study design, data collection and analysis, decision to publish, or preparation of the manuscript.

**Competing interests:** Dr. MDP, M.D., is a Co-Founder and Chief Scientific Officer (CSO) of MicroRid Technologies Inc. The goal of MicroRid Technologies Inc. is to develop new antifungal agents for therapeutic use. All other authors declare no competing interests.

pathogenicity. Overall, this study demonstrates that single allele dysfunction of *ERG251* is a recurrent and effective mechanism of acquired azole tolerance. We propose that altered sterol composition resulting from *ERG251* dysfunction mediates azole tolerance as well as pleiotropic effects on stress response, filamentation and virulence.

## Author summary

Invasive infections caused by the fungal pathogen *Candida albicans* have high mortality rates (20–60%), even with antifungal drug treatment. Numerous mechanisms contributing to drug resistance have been characterized, but treatment failure remains a problem indicating that there are many facets that are not yet understood. The azole class of antifungals target production of ergosterol, an essential component of fungal cell membranes. Here, we provide insights into how *ERG251*, a component of the ergosterol biosynthesis pathway, contributes to enhanced growth in azoles, along with broader impacts on stress responses, filamentation, and pathogenicity. One of the most striking results from our study is that even a single nucleotide change in one allele of *ERG251* in the diploid *C. albicans* can lead to azole tolerance. Tolerance, a distinct phenotype from resistance, is the ability of fungal cells to grow above the minimum inhibitory concentration in a drug concentration-independent manner. Tolerance frequently goes undetected in the clinic because it is not observable in standard assays. Strikingly, azole tolerant strains lacking one allele of *ERG251* remained virulent in a mouse model of infection highlighting the potential for mutations in *ERG251* to arise and contribute to treatment failure in patients.

## Introduction

*Candida albicans* is the most prevalent human fungal pathogen, affecting millions of people and leading to severe and potentially fatal infections, particularly in individuals with weakened or compromised immune systems [1–4]. Invasive infections caused by *C. albicans* can result in mortality rates nearing ~60% despite the existing antifungal treatments [1,2,5,6]. Treatment failures and infection recurrences are common [1,2,7,8]. One contribution to treatment failure is drug resistance, which is defined as the ability to grow above the minimum inhibitory concentration (MIC) of a drug-susceptible isolate at rates similar to growth in the absence of drug. However, treatment failure can also occur in strains that are classified as susceptible based on MIC. This highlights the importance of drug tolerance, which is the ability of a fungus to grow slowly above the MIC in a drug concentration-independent manner [7,9,10].

Azole antifungal drugs target the biosynthesis of ergosterol which is an essential component of fungal cell membranes and acts to maintain cell membrane integrity and fluidity. Azole exposure leads to the depletion of ergosterol and accumulation of a toxic sterol 14-α-methylergosta-8,24(28)-dien-3β,6α-diol (herein referred to as 'toxic dienol') that permeabilizes the plasma membrane, arrests fungal growth, and increases sensitivity to environmental stresses [11–13]. During treatment with fungistatic azoles, many *Candida* species can rapidly evolve drug resistance through various mechanisms including modification or overexpression of the gene encoding the drug target *ERG11* and upregulation of drug efflux pumps encoded by *CDR1*, *CDR2*, and *MDR1* [14,15]. However, these are not the only possible mechanisms. For example, the transcription factor Adr1 has recently been identified as a key regulator of ergosterol biosynthesis, and hyperactivation of Adr1 confers azole resistance in *C. albicans* [16].

Ergosterol biosynthesis is broadly conserved among Saccharomycotina which includes *Candida* species as well as the baker's yeast *Saccharomyces cerevisiae*. However, some differences in gene duplication and expression patterns in the more than 20 enzymes along the ergosterol biosynthetic pathway have been identified [17–20]. Ergosterol biosynthesis is divided into three parts: the mevalonate, late, and alternate pathways [19,21]. The mevalonate pathway is responsible for the production of farnesyl diphosphate (FPP), an important ergosterol intermediate. Dephosphorylation of FPP generates farnesol, a quorum-sensing molecule that can regulate the yeast-to-hyphae transition and biofilm formation in *C. albicans* [22–24]. The late pathway is responsible for using FPP to synthesize ergosterol. The rate-limiting enzyme, lanosterol 14-α-demethylase, Erg11 is the direct target of azoles. Inhibition of Erg11 by azoles decreases the production of ergosterol, which negatively affects the cell, and results in the accumulation of its substrate lanosterol that feeds into the alternate pathway and proceeds toward the production of the toxic dienol. Key enzymes in the alternate pathway, Erg6 and Erg3, are respectively responsible for the initial step and last step of toxic dienol generation [11,19,25]. Inactivation or modification of *ERG3* or *ERG6* impacts drug susceptibility of many *Candida* species [11,18,20,26–31]. For example, loss of Erg6 function reduces susceptibility to nystatin and polyenes in *C. glabrata* [26,32,33]. Loss of Erg3 function confers resistance to azoles in *C. albicans*, *C. parapsilosis*, *C. dubliniensis* and resistance to polyenes in *C. albicans* and *C. lusitaniae* [20,27–29,34,35]. *ERG3* inactivation causes reduced toxic dienol and instead results in accumulation of 14α-methylfecosterol which supports growth in the presence of azoles despite altered membrane composition [11,20,25,36–38]. *ERG3* inactivation also rescues the lethality of *ERG11* deletion mutants in multiple species [11,20,25,36–38].

Both the late and alternate pathways of ergosterol biosynthesis utilize C-4 sterol methyl oxidase, the catalytic component of the C-4 demethylation complex that is responsible for removing the two methyl groups from the C-4 position of the sterol molecule [18,19,39]. In *Saccharomyces cerevisiae*, Erg25 is the solo C-4 sterol methyl oxidase and essential for standard growth [19,40]. However, both *Aspergillus fumigatus* and *C. albicans* encode two membrane C-4 sterol methyl oxidases with one of them serving as the primary enzyme during biosynthesis [17,18,39]. In *C. albicans*, *ERG251* encodes the primary C-4 sterol methyl oxidase, but few studies have characterized it independently and the findings are contradictory. For example, *erg251Δ/Δ* exhibited increased fluconazole (FLC) susceptibility and accumulation of eburicol, the direct precursor for 14α-methylfecosterol, in the presence of FLC [18]. However, in a haploid *C. albicans* strain, transposon insertion into *ERG251* resulted in decreased FLC susceptibility [34]. This contradiction highlights the need to understand the effect of growth conditions and genetic background on the relationship between *ERG251* and drug susceptibility.

Proper levels of ergosterol are crucial for multiple cellular functions including stress response, nutrient transport, and host-pathogen interactions [19,39]. Deletion or overexpression of the key genes in the ergosterol biosynthetic pathway disrupts ergosterol biosynthesis and results in increased susceptibility to osmotic and cell wall stress [19,41]. Disruption of *ERG6* and *ERG24* also leads to the reduced transport of potassium, calcium and metal in *S. cerevisiae* and *A. fumigatus* [42–44]. Furthermore, *C. albicans erg2Δ/Δ* and *erg24Δ/Δ* mutants exhibit abnormal vacuolar physiology and filamentation defects, and are avirulent in a disseminated model of candidiasis [45].

In this report, we determined the effects of heterozygous and homozygous inactivation of *ERG251* on drug susceptibility, sterol composition, filamentation, virulence, and response to stress. Strikingly, we identified recurrent heterozygous point mutations in *ERG251* in two distinct genetic backgrounds (SC5314 and P75063) in three independent FLC evolution experiments. Azole tolerance occurred with single allele dysfunction of *ERG251* in both of the

euploid genetic backgrounds. Azole resistance occurred with single allele dysfunction of *ERG251* in combination with concurrent aneuploidy of chromosome 3 and chromosome 6. Homozygous deletion of *ERG251* resulted in increased fitness in the presence of low concentrations of FLC, but decreased fitness in rich medium, especially at low cell density. In the presence of FLC, all *ERG251* deletion mutants (heterozygous and homozygous) had decreased accumulation of the toxic dienol and increased accumulation of non-toxic alternative sterols. We conclude that *ERG251* is the solo active C-4 sterol methyl oxidase of the alternative sterol pathway in *C. albicans* and dysfunction of this enzyme promotes survival in the presence of azole drugs due to the accumulation of non-toxic alternative sterols. Changes in sterol composition also support the pleiotropic effects of *ERG251* on transcription, filamentation, and stress responses. Lastly, the *erg251Δ/Δ* mutant had decreased virulence while both heterozygous deletion mutants maintained their pathogenicity. In summary, we demonstrate a novel mechanism of Erg251-mediated azole drug tolerance via an accumulation of non-toxic sterol intermediates and highlight the complex and pleiotropic effects of these changes on fitness, stress responses, filamentation, and pathogenicity.

## Results

### Recurrent point mutations in *ERG251* evolve during adaptation to fluconazole

During *in vitro* evolution of *C. albicans* in the presence of FLC, *ERG251* point mutations were recurrently detected in three independent experiments and within two distinct genetic backgrounds: P75063 and SC5314 (Table 1) [10]. Using whole genome sequencing (WGS), we detected *ERG251* point mutations in 16 FLC-evolved strains (Table 1). In P75063 *ERG251* is homozygous, while in the type strain SC5314, there are two non-synonymous variants between *ERG251-A* and *ERG251-B*, and *de novo* point mutations were identified in both alleles. The point mutations were all heterozygous and were characterized as missense (A60D, A268D,

**Table 1. List of FLC-evolved strains with different *ERG251* point mutations.** *ERG251* mutations were identified in three independent FLC evolution experiments within two distinct genetic backgrounds (SC5314-derived Sn152 and BWP17, and P75063). Multiple single-colony strains from the same evolved population are organized into the same row. *ERG251* point mutations are annotated on the mutated allele, and represented as allele A/allele B, except for AMS5615 where the chromosome region containing *ERG251* became homozygous prior to acquisition of the point mutation. *ERG251* in the P75063 background is homozygous.

| Strains | Genetic background | *ERG251* genotypes | Mutation types | Copy number changes | FLC MIC (24hr) | FLC SMG (48hr) |
|---|---|---|---|---|---|---|
| \multicolumn{7}{c}{Strains from evolution experiment 1 [Zhou et al., 2024, accepted]} |||||||
| SN152 | N.A | *ERG251/ERG251* | N.A | None | 0.5 μg/ml | 0.07 |
| Evolved 1.1/1.2 | Sn152 | *ERG251/ERG251$^{L113*}$* | Stop gained | Chr3x3, Chr6x3, Chr5 LOH | >256 μg/ml | N. A |
| Evolved 2.1/2.2 | Sn152 | *ERG251$^{E273*}$/ERG251* | Stop gained | Chr3x3, Chr6x3 | >256 μg/ml | N. A |
| Evolved 3.1/3.3 | Sn152 | *ERG251$^{A60D}$/ERG251* | Missense | Chr3x3, Chr6x3 | >256 μg/ml | N. A |
| Evolved 3.2 | Sn152 | *ERG251/ERG251$^{*322Y}$* | Stop lost | Chr6x3 | 1 μg/ml | 0.58 |
| \multicolumn{7}{c}{Strains from evolution experiment 2} |||||||
| BWP17 | N.A | *ERG251/ERG251* | N.A | None | 0.5 μg/ml | 0.08 |
| AMS5615 | BWP17 | *ERG251/ERG251$^{A268D}$* | Missense | Chr4 (partial)x4, Chr7x3, Chr7 LOH | 1 μg/ml | 0.79 |
| AMS5617/5618 | BWP17 | *ERG251$^{S27fs}$/ERG251* | Frame shift | Chr4x3,Chr5 LOH | 1 μg/ml | 0.73 |
| AMS5622/5623/5624 | BWP17 | *ERG251$^{G62W}$/ERG251* | Missense | Chr7x3 | 1 μg/ml | 0.54 |
| AMS5625/5626 | BWP17 | *ERG251$^{W265G}$/ERG251* | Missense | None | 1 μg/ml | 0.50 |
| \multicolumn{7}{c}{Strains from evolution experiment 3 [10]} |||||||
| P75063 | N.A | *ERG251/ERG251* | N.A | None | 0.5 μg/ml | 0.11 |
| AMS4130 | P75063 | *ERG251/ERG251$^{H274Q}$* | Missense | None | 1 μg/ml | 0.26 |

G62W, W265G, and H274Q), nonsense stop gained (L113* and E273*), frameshift (S27fs), or stop lost (*322Y) (Table 1). Many of these *ERG251* point mutations arose in evolved strains with large genomic copy number changes, predominantly whole chromosome trisomies (Table 1 and S1A Fig). The high frequency of mutations in *ERG251* during exposure to antifungal drug suggests an important role of *ERG251* in response to antifungal drug stress.

## Single allele dysfunction of *ERG251* leads to azole drug tolerance

To determine the impact of *ERG251* point mutations on drug susceptibility, we quantified azole resistance and tolerance in the evolved isolates. All FLC-evolved strains carrying *ERG251* point mutations in the SC5314-derived background were either resistant (minimal inhibitory concentration, MIC$\geq$ 256μg/ml) or tolerant (Supra-MIC growth, SMG >0.50) to FLC and other azoles (voriconazole, VOC, and itraconazole, ITC) (Tables 1 and S1). Similarly, the FLC-evolved strain carrying an *ERG251* point mutation in the P75063 background had increased tolerance relative to the progenitor (increase in SMG from 0.13 to 0.26) (Tables 1 and S1).

Next, we engineered representative point mutations from the FLC-evolved strains into the wild-type, drug-sensitive SC5314 background and determined azole susceptibility. Four different heterozygous *ERG251* point mutations (L113*, W265G, E273*, and *322Y) were selected to represent the range of drug tolerant and resistant phenotypes. The engineered point mutants resulted in a 2-fold increase in MIC and more than an 8-fold increase in tolerance to three different azole drugs (FLC, VOC, ITC) (Fig 1A and S1 Table).

Based on these phenotypes we hypothesized that the heterozygous point mutations were due to a loss of gene function. We generated two different heterozygous deletion mutants of *ERG251* by deleting either the A or the B allele in the SC5314 background (aΔ/B: *erg251Δ/ERG251* and A/bΔ: *ERG251/erg251Δ*). Additionally, we constructed a strain with heterozygous over-expression of *ERG251* (*ERG251*/TetO-*ERG251*) in the SC5314 background. We validated the deletion mutants using WGS and confirmed that transformation did not introduce off-target effects (S1B Fig). Heterozygous deletion of *ERG251* resulted in azole tolerance levels that were the same as all four engineered strains with heterozygous *ERG251* point mutations (Fig 1A and S1 Table). Meanwhile, over-expression of *ERG251* only resulted in a small increase in FLC tolerance (SMG = 0.3, Fig 1B), but did not phenocopy the evolved mutants. Complementation of the heterozygous mutants *erg251Δ/ERG251* and *ERG251/erg251Δ* with the missing *ERG251* allele (*erg251Δ/ERG251+ERG251-A* and *ERG251/erg251Δ+ERG251-B*) eliminated the FLC tolerance (Figs 1B and S1B). In the P75063 genetic background, heterozygous deletion of *ERG251* was sufficient to cause the increase in azole tolerance observed for the FLC-evolved strain that carried the *ERG251* point mutation (Fig 1B and Table 1). Therefore, we conclude that these *ERG251* point mutations lead to the single allele dysfunction of *ERG251* which causes azole tolerance in *C. albicans*.

Next, we tested if *ERG251*-mediated azole tolerance was caused by upregulated drug efflux pumps or dependent on Hsp90. Measurement of Rhodamine 6G (R6G) is a useful method for quantifying efflux pump activity [46]. We found that *ERG251*-mediated azole tolerance was independent of drug efflux pumps as indicated by a small decrease in the rate of efflux of R6G for *ERG251* heterozygous deletion mutants compared to *ERG251/ERG251* (SC5314) during the exposure to FLC (Fig 1C). Hsp90 is a molecular chaperone and an important mediator for drug tolerance and stress response [47,48]. We found that *ERG251*-mediated tolerance depends on Hsp90 function. Addition of an Hsp90 inhibitor (radicicol, 2.5μM) to assays measuring azole resistance (MIC$_{50}$) and tolerance (SMG) blocked the acquired azole tolerance of *ERG251* heterozygous deletion mutants. Radicicol did not reduce the MIC or the cell viability of a well-characterized FLC resistant clinical isolate with increased expression of *ERG11*,

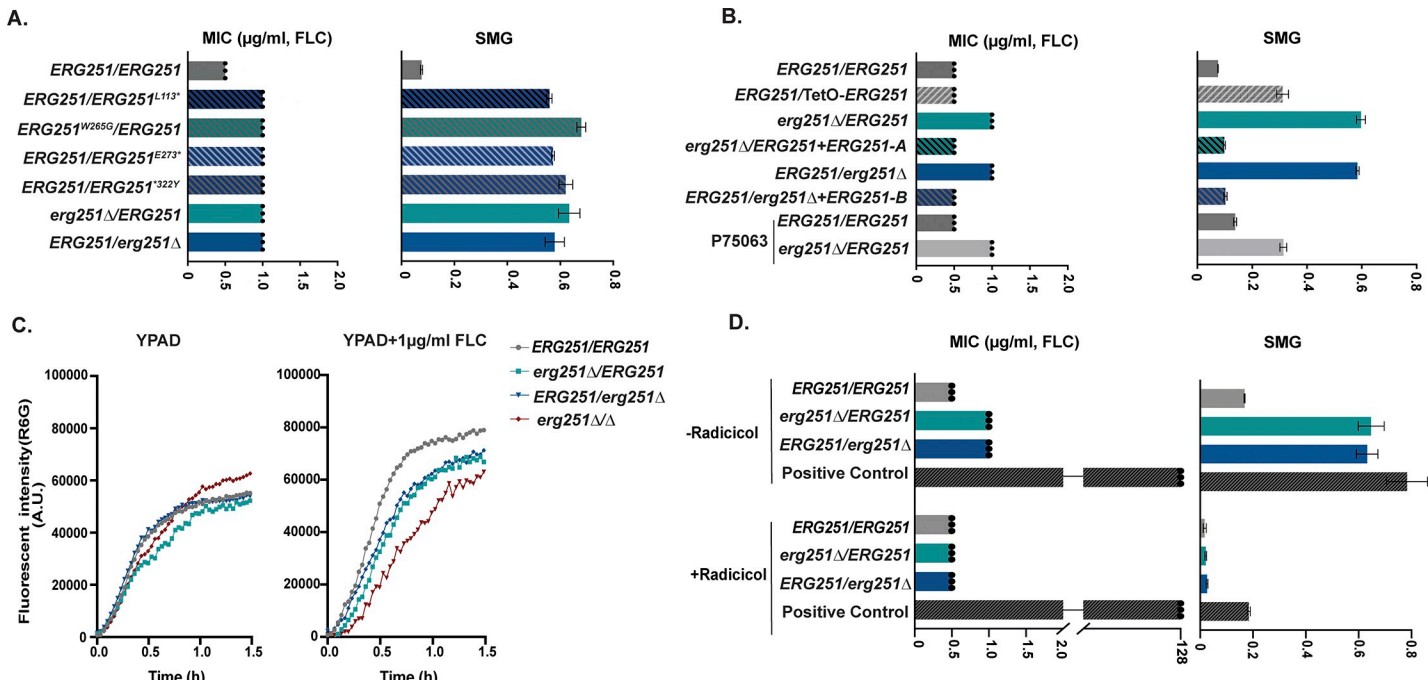

**Fig 1. The point mutation of *ERG251* leads to the partial dysfunction of *ERG251* causing acquisition of azole tolerance.** Liquid microbroth drug susceptibility assay. Fluconazole (FLC) resistance quantified as the $MIC_{50}$ at 24hr in increasing concentrations of FLC (left) and FLC tolerance quantified as the Supra-MIC growth at 48hr (SMG, right) which is the average growth above the $MIC_{50}$ for: **A.** the wild-type SC5314 (*ERG251/ERG251*), engineered heterozygous *ERG251* point mutations strains in the SC5314 background, and both heterozygous deletion mutants of *ERG251* in the SC5314 background; and **B**. the wild-type SC5314 (*ERG251/ERG251*), *ERG251* overexpression strain, both heterozygous deletion mutants of *ERG251* and their corresponding complementation strains, an *ERG251* heterozygous deletion in the P75063 background and wild-type P75063 (P75063-*ERG251/ERG251*) as a control. **C.** Rhodamine 6G efflux kinetics of two heterozygous deletion mutants and the homozygous deletion of *ERG251* in SC5314 with SC5314 (*ERG251/ERG251*) as the control in YPAD (left) and YPAD+1µg/ml FLC (right). Plots indicate average fluorescence intensity changes of Rhodamine 6G (R6G) from three biological replicates over 90 min. **D.** 24hr MIC (left, µg/ml) and 48hr SMG (right, tolerance) in FLC with or without radicicol (Hsp90 inhibitor) treatment for two heterozygous deletion mutants of *ERG251* with SC5314 (*ERG251/ERG251*) and a Positive Control (a FLC resistant clinical isolate (C17/12-99)). **A&B&D:** For MIC values, each dot represents a single replicate and each bar represents the average of three biological replicates of a single strain; SMG values are mean ± SEM calculated from three biological replicates of a single strain.

*MDR1*, *CDR1* and *CDR2* (C17/12-99, Fig 1D and S1 Table) [49,50]. Additionally, for susceptible and tolerant strains, inhibition of Hsp90 caused FLC to become fungicidal as no viable cells were recovered from higher FLC concentrations combined with radicicol (S2 Fig). Hsp90 regulates cell morphogenesis and cell wall stress through the calcineurin pathway, suggesting that *ERG251*-mediated tolerance may also alter cell membrane and/or cell wall stress responses [48].

## Single allele dysfunction of *ERG251* and concurrent aneuploidy leads to azole resistance

The single allele dysfunction of *ERG251* was sufficient to reproduce the azole tolerance phenotype observed in 10/16 of the FLC-evolved strains with *ERG251* point mutations. However, 6 of the FLC-evolved strains with *ERG251* point mutations acquired *bona fide* azole resistance (MIC >256 µg/ml FLC, Table 1). All 6 resistant *ERG251* mutants also had chromosome (Chr) 3 and Chr6 concurrent aneuploidies (Fig 2A and Table 1, Evolved 1.1/1.2, 2.1/2.2 and 3.1/3.3 representing two single colonies from three independent FLC-evolved lineages). Recently, we showed that Chr3 and Chr6 concurrent aneuploidy causes azole tolerance and correlates with elevated expression of drug responsive genes located on these chromosomes, including *CDR1*, *CDR2*, *MDR1*, and *MRR1* [Zhou et al. 2024, accepted]. We hypothesized that heterozygous

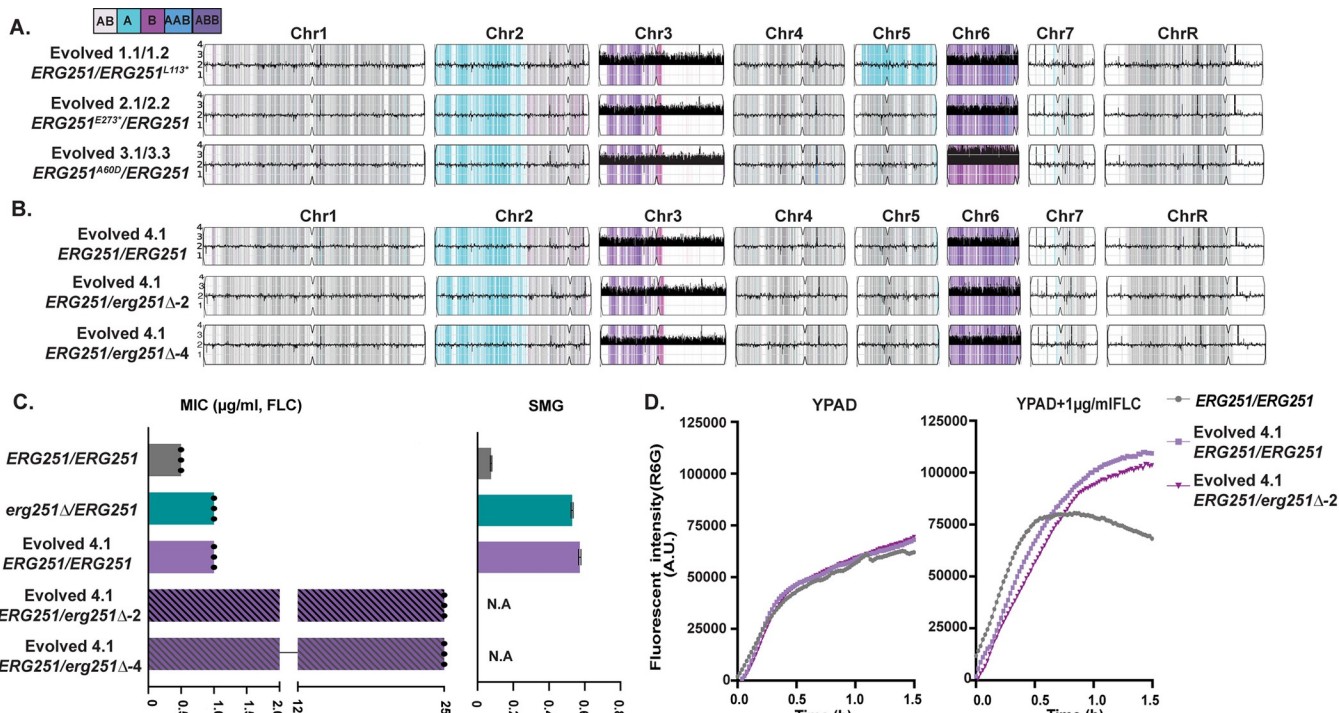

**Fig 2. Single allele dysfunction of *ERG251* in combination with concurrent aneuploidy causes azole resistance. A.** Representative whole genome sequencing (WGS) data of the FLC-evolved strains 1.1/1.2, 2.1/2.2, and 3.1/3.2 that acquired heterozygous point mutations at *ERG251* and Chr3 and Chr6 concurrent aneuploidy. **B.** WGS data of FLC-evolved strain 4.1 that had wild-type alleles of *ERG251/ERG251* and Chr3 and Chr6 concurrent aneuploidy, plus two *ERG251* heterozygous deletion mutants engineered in the Evolved 4.1 aneuploid background. **A&B** WGS data are plotted as the log2 ratio and converted to chromosome copy number (y-axis, 1–4 copies) as a function of chromosome position (x-axis, Chr1-ChrR). The baseline ploidy was determined by propidium iodide staining (S1 Table). Haplotypes relative to the reference genome SC5314 are indicated. **C.** 24hr MIC (left, µg/ml) and 48hr SMG (right, tolerance) in FLC for SC5314 (*ERG251/ERG251*), *ERG251* heterozygous deletion mutant in the SC5314 background, FLC-evolved strain 4.1, and two *ERG251* heterozygous deletion mutants engineered in the Evolved 4.1 aneuploid background (two independent transformants). MIC: each dot represents a single replicate and bar represents the average of three technical replicates of a single strain; SMG values are mean ± SEM calculated from three technical replicates of a single strain. **D.** Rhodamine 6G efflux kinetics of *ERG251* heterozygous deletion mutant in evolved strain 4.1 background with evolved strain 4.1 and SC5314 (*ERG251/ERG251*) as the controls in YPAD (left) and YPAD+1µg/ml FLC (right). Plots indicate fluorescence intensity changes of Rhodamine 6G (R6G) over 90 min.

deletion of *ERG251* in the Chr3 and Chr6 aneuploid background would make these tolerant cells resistant. To test this hypothesis, we isolated an azole tolerant strain with Chr3 and Chr6 concurrent aneuploidies and wild-type alleles of *ERG251* in the SC5314-derived genetic background (Evolved 4.1: *ERG251/ERG251*) (Fig 2B). We deleted one copy of *ERG251* (on Chr4) from this concurrent aneuploid strain, and confirmed that the mutants maintained the aneuploid chromosomes by whole genome sequencing. Heterozygous deletion of *ERG251* in the concurrent aneuploidy background with elevated drug efflux resulted in a 256-fold increase in MIC, reproducing the azole resistance phenotype observed for the FLC-evolved resistant strains (>256µg/ml, Fig 2B, 2C, and 2D and S1 Table). We therefore conclude that the combination of the single allele dysfunction of *ERG251* and concurrent aneuploidy leads to *bona fide* drug resistance.

## Erg251 exhibits contrasting effects on fitness in the presence or absence of drug

Our results show that disruption of one copy of *ERG251* results in tolerance or resistance to azoles in distinct genetic backgrounds. Although we have shown that Hsp90 is required for tolerance, the full range of mechanisms and impact of changing *ERG251* are not known. In order

to fully understand the function of *ERG251* in different cellular processes, two independent homozygous *ERG251* deletion mutants were generated in SC5314 background. We confirmed that these deletions did not introduce any large-scale genomic changes (loss of heterozygosity or aneuploidy) (S1B Fig), and two independent *erg251Δ/Δ* mutants (d51 and d70) exhibited identical phenotypes. Both e*rg251Δ/Δ* mutants had decreased growth in rich medium with low initial cell density ($OD_{600}$ = 0.001) compared to the wild-type control in a 96-well plate format with constant shaking (Fig 3A and 3B). Importantly, we found that the growth defect of *erg251Δ/Δ* in YPAD could be partially rescued by simply increasing initial cell density (from an $OD_{600}$ = 0.001 to 0.005 or 0.01) (Fig 3A and 3B). In *C. albicans*, cell density is communicated and linked with gene expression via the quorum sensing process, and farnesol is a major quorum sensing molecule secreted by *C. albicans* [22,51,52]. The production of farnesol requires the dephosphorylation of FPP, the precursor for the ergosterol biosynthesis pathway [23,53]. Therefore, we tested the impact of different concentrations of farnesol (0–1000μM) on the growth of *erg251Δ/Δ* mutants with low initial cell density (Fig 3C, Y-axis). Moderate concentrations of farnesol (62.5–250 μM) improved the growth of *erg251Δ/Δ* in YPAD, while farnesol had no impact on growth of the wild-type control (Fig 3D, Y-axis). Therefore, homozygous deletion of *ERG251* may result in disrupted ergosterol biosynthesis which subsequently provides negative feedback on farnesol production contributing to the growth defect of *erg251Δ/Δ*.

We next measured the impact of FLC on *erg251Δ/Δ* strains. The growth defect of the *erg251Δ/Δ* strain prevented us from conducting MIC and SMG assays for resistance and tolerance because these assays are normalized to growth in rich media (no drug) and *erg251Δ/Δ* strains grow poorly in these conditions. Therefore, we tested the impact of different concentrations of FLC (0–256μg/ml) on *erg251Δ/Δ* using a growth curve assay. Low concentrations of FLC (≤1μg/ml) increased growth of *erg251Δ/Δ* compared to no drug, whereas higher concentrations of FLC had no impact and growth remained poor (Fig 3C, X-axis). In contrast, the wild-type control (*ERG251/ERG251*) exhibited decreased growth at concentrations at and above its $MIC_{50}$ (0.5μg/ml FLC, Fig 3D, X-axis). This suggests that total dysfunction of *ERG251* can promote *C. albicans* growth in FLC but only at low concentrations.

Adding either farnesol or FLC only partially restored the growth defect of *erg251Δ/Δ* (Fig 3C). Therefore, we determined if adding farnesol in combination with FLC could further restore growth of *erg251Δ/Δ*. Growth of *erg251Δ/Δ* from all different concentration combinations showed that low concentrations of FLC (≤1μg/ml) are sufficient to confer increased growth regardless of the concentration of farnesol (Fig 3C). In contrast, high concentrations of both farnesol (>125μM) and FLC (>1μg/ml) greatly inhibited growth of *erg251Δ/Δ* (Fig 3C). Growth inhibition for the wild-type control (*ERG251/ERG215*) was solely controlled by the FLC concentration (Fig 3D). Furthermore, high-concentration farnesol (>125μM) combined with high concentrations of FLC (>64μg/ml) exhibited a killing effect on the *erg251Δ/Δ* cells but not on the wild-type control (*ERG251/ERG251*) (Fig 3E and 3F). These results suggest that in the absence of Erg251, farnesol can make FLC fungicidal at high concentrations, likely due to more severe inhibition of cell growth and ergosterol production.

Lastly, a head-to-head competition assay validated the fitness trade-off for *erg251Δ/Δ*, with a fitness cost in YPAD and a fitness benefit in the presence of a low concentration of FLC (1μg/ml) (Fig 3G). This fitness trade-off was not seen for the two heterozygous deletion mutants (*erg251Δ/ERG251* or *ERG251/erg251Δ*) and was completely rescued by complementation of the homozygous deletion mutant with either the *ERG251-A* or *ERG251-B* allele (*erg251Δ/Δ+ERG251-A* or *erg251Δ/Δ+ERG251-B*) (Fig 3G). Taken together, we propose that in response to low concentrations of FLC, *erg251Δ/Δ* upregulates the alternate sterol production pathway to compensate for a reduction in ergosterol production and support increased growth (see below).

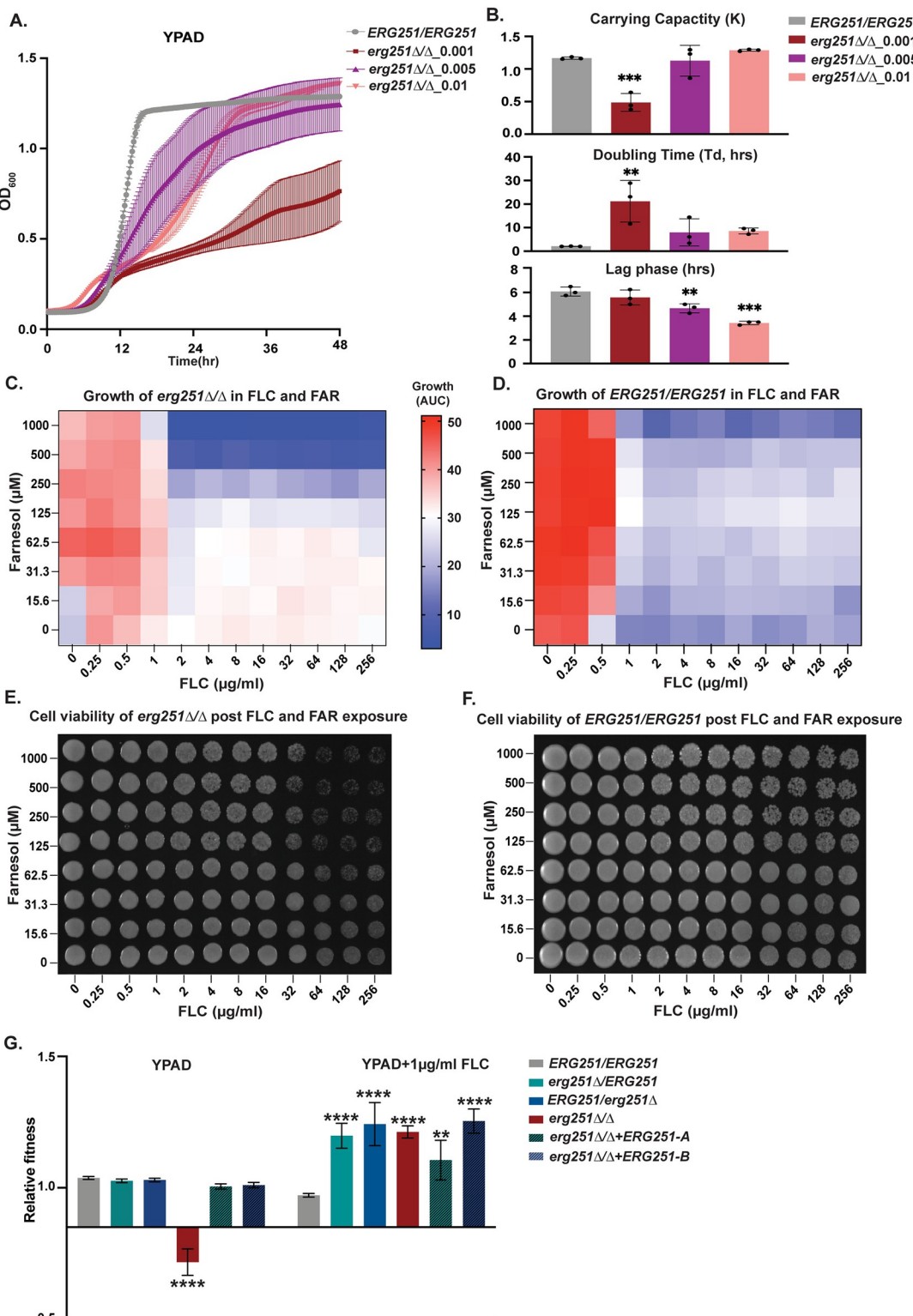

**Fig 3. Homozygous deletion of *ERG251* results in decreased fitness at low initial cell density and increased fitness in the presence of low concentrations of FLC (≤1μg/ml). A.** 48hr growth curve analysis of *erg251Δ/Δ* started at three different initial cell densities (OD$_{600}$ = 0.001, 0.005, or 0.01) with *ERG251/ERG251* (SC5314, OD$_{600}$ = 0.001) as the control. Average slope and ±SEM for three technical replicates is indicated. **B.** Carrying capacity (K) and doubling time (Td, hrs), and lag phase (hrs) determined from growth curve analysis in Fig 2A. **C&D.** X-Y growth curve assay of **(C)** *erg251Δ/Δ* and **(D)** *ERG251/ERG251* in

the presence of increasing concentrations of FLC (X-axis, 0–256 µg/ml, 2-fold dilutions) and/or increasing concentrations of farnesol (FAR) (Y-axis, 0–1000 µM, 2-fold dilutions). Growth was estimated with the area under the curve (AUC heatmap) of the 48hr growth curve. **E&F.** Cell viability of (**E**) *erg251Δ/Δ* and (**F**) *ERG251/ERG251* after 48 hr exposure to FLC or/and FAR. Cells from Fig 3B were plated on YPAD agar and imaged after 24hr incubation. **G.** Relative fitness calculated from head-to-head competitive assay for *erg251Δ/ERG251*, *ERG251/erg251Δ*, *erg251Δ/Δ*, *erg251Δ/Δ+ERG251-A*, and *erg251Δ/Δ+ERG251-B* compared to the fluorescent control strain (*ERG251/ERG251*). **B&G**: Values are mean ± SEM calculated from three technical replicates. Data were assessed for normality by Shapiro-Wilk, and significant differences between the *ERG251/ERG251* and mutants were calculated using two-way ANOVA with Dunnett's multiple comparisons test. ****$p<0.0001$, **$p<0.01$. **A-G:** At least three biological replicates were performed.

## Pleiotropic effects of Erg251 on cell membrane organization, stress response, and biofilm formation

We next explored the mechanisms by which *ERG251* affects fitness, drug susceptibility, and other stress responses. Transcriptional analysis was performed for the SC5314 wild-type (*ERG251/ERG251)*, two *ERG251* heterozygous deletion mutants, and one homozygous deletion mutant using RNAseq in two different log phase conditions: YPAD and YPAD+1µg/ml FLC. We first focused our analysis on the comparison between *erg251Δ/Δ* and wild-type in YPAD to understand the role of *ERG251* in a broad range of cellular processes (Fig 4A and S2 Table). Differential expression analysis was used to identify genes with a significant change in abundance in *erg251Δ/Δ* cells compared to wild-type (913 genes, log2 fold change $\geq$ 1 or $\leq$-1 and adjusted p-value < 0.05). Gene Ontology (GO) analyses of differentially expressed genes revealed an overrepresentation of genes associated with cell wall organization, biofilm formation, filamentation growth, metabolic processes, and stress response (Fig 4B and S3 Table).

We identified many down-regulated genes in *erg251Δ/Δ* cells involved in cell membrane, filamentation, and stress response. Genes that regulate cell membrane structure including lipid metabolism, ergosterol and sphingolipid biosynthesis were down-regulated, including *ERG11*, *ERG1*, *LIP1*, *PLB1*, *UPC2*, *ARE2*, and *SCS7* (Fig 4A and 4B) [34,54]. Genes that regulate filamentation and biofilm formation were also down-regulated, including *FGR23*, *HYR1*, *ALS2*, and *ALS4* (Fig 4A and 4B) [55–57]. Additionally, the osmotic stress related gene *AQY1* and heat stress related genes *HSP70* and *HSP12* were also down-regulated (Fig 4A) [57–59]. In contrast, genes that are involved in oxidative stress response like *SOD5* and *AOX2* were up-regulated (Fig 4A and 4B) [60,61]. These changes in lipid metabolism and stress response may affect metabolism and nutrient availability more broadly including carbon and amino acid metabolism [18,62].

GO analysis of biological process identified enrichment of 26 genes in *erg251Δ/Δ* cells that encode proteins with GlycosylPhosphatidylInositol (GPI)-anchored motifs. GPI anchors attach proteins to the cell surface contributing to cell-wall integrity, cell-cell interaction, and hyphal formation [63,64]. These genes include *HYR1*, *FGR23*, and *SOD5* as well as cell wall specific genes in the *PGA* and *ALS* families. Overall, genes encoding GPI-anchored motifs were down-regulated in *erg251Δ/Δ* cells (20 out of 26) (Fig 4A and S4 Table) consistent with prior work demonstrating cross-talk between the ergosterol and GPI biosynthesis pathways [65,66].

Phenotypic analysis was consistent with transcriptional analysis for the pleiotropic effects of *ERG251* on cell membrane organization and stress responses. Compared to wild-type (*ERG251/ERG251*), *erg251Δ/Δ* exhibited no change in response to increased temperature (37˚C) or cell wall stressors (Calcofluor White and Congo Red) (S2B Fig). In contrast, *erg251Δ/Δ* exhibited detectable phenotypes in response to cell membrane, osmotic and oxidative stress (S2B Fig). To quantify these effects, we performed growth curve analysis in the absence and presence of increasing concentrations of $H_2O_2$, NaCl, and SDS, and calculated the minimum concentration that inhibited growth by 20% relative to no stress. The *erg251Δ/Δ*

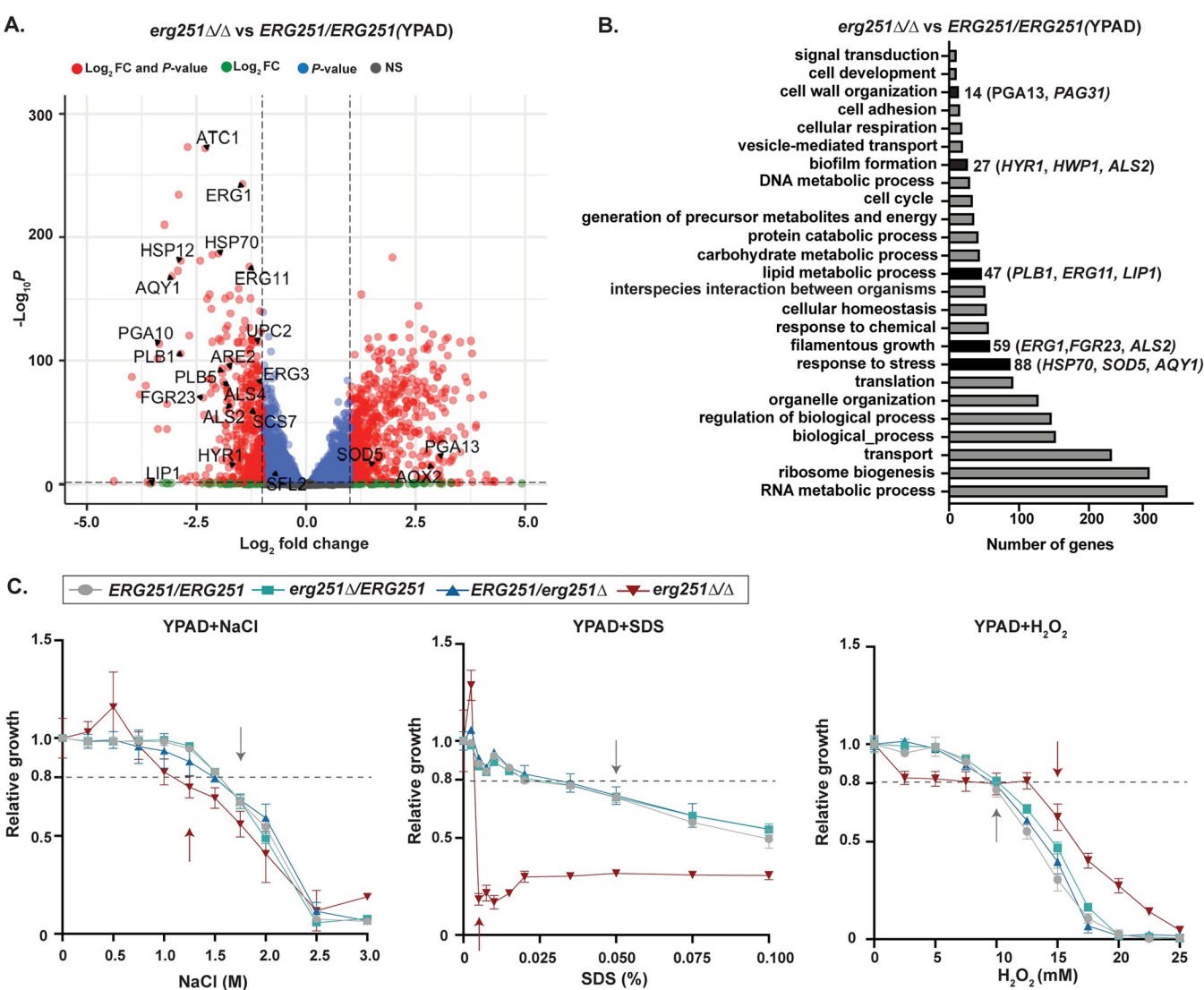

**Fig 4. Homozygous deletion of *ERG251* leads to increased sensitivity to cell membrane and osmotic stress but decreased sensitivity to oxidative stress. A.** Volcano plot for differentially expressed genes (log$_2$ fold change $\geq$ 1 or $\leq$-1 and adjusted *p*-value < 0.05) in the *erg251$\Delta$/$\Delta$* mutant compared to *ERG251/ERG251* in YPAD. Genes that are significantly differentially expressed by both fold change and p-value cut-offs are in red. **B**. Gene Ontology (GO) terms for differentially expressed genes (log$_2$ fold change $\geq$ 1 or $\leq$ -1 and adjusted *p*-value < 0.05) in the *erg251$\Delta$/$\Delta$* mutant compared to *ERG251/ERG251* in YPAD. Cell wall organization, biofilm formation, lipid metabolic process, filamentous growth and response to stress. GO terms are highlighted and differentially expressed genes contributing to the enrichment noted to the right. **C.** Relative growth (area under growth curve) of *ERG251/ERG251*, *erg251$\Delta$/ERG251*, *ERG251/erg251$\Delta$*, and *erg251$\Delta$/$\Delta$* in YPAD+NaCl (0 to 3.0M), YPAD+SDS (0 to 0.1%), and YPAD+H$_2$O$_2$ (0 to 25 mM) across different concentrations (Methods). Relative growth was calculated by normalizing to the growth of no drug control. Arrows indicate the minimum concentration that inhibits the growth (20%) of *ERG251/ERG251* (gray) and *erg251$\Delta$/$\Delta$* (red) relative to no drug control. Dashed line indicates the cut-off for the 20% decreased growth. Data are presented as the mean ±SEM for three technical replicates. **A-C**: At least three biological replicates were performed.

mutant was more susceptible to osmotic (NaCl, 1.4-fold decrease) and cell membrane (SDS, 10-fold decrease) stress, but exhibited increased resistance to H$_2$O$_2$ (1.5-fold increase) relative to wild-type (Figs 4C and S2B). These changes in stress response were not observed for the two *ERG251* heterozygous deletion mutants at either the transcriptional or phenotypic levels (Figs 4C and S3A–S3D). Taken together, this indicates that the total loss of *ERG251* results in a dramatic physiological response that impacts cell membrane composition and osmotic/oxidative stress responses.

### *ERG251-A* exhibits dominant regulation of filamentation

Given that genes related to filamentation and biofilm formation were downregulated in *erg251Δ/Δ*, we next quantified filamentation in all three deletion mutants (homozygous and two heterozygous). Using an *in vitro* filamentation assay, we found that deletion of *ERG251-A* (*erg251Δ/ERG251*) had a ~25% decrease in the proportion of hyphae, while deletion of *ERG251-B* (*ERG251/erg251Δ*) exhibited no change compared to wild-type *ERG251/ERG251* (Fig 5A and 5B). Complementation of *erg251Δ/ERG251* with the *ERG251-A* allele restored wild-type filamentation (Fig 5A and 5B). Similar filamentation defects were also observed for the FLC-evolved strains with *ERG251* loss-of-function point mutations in the A allele (*ERG251^{W265G}/ERG251*), but not for evolved strains with point mutations in the B allele (*ERG251/ERG251^{*322Y}*) (S3E Fig). This indicates that *ERG251-A* plays a dominant role in regulating filamentation, while *ERG251-B* is not required for filamentation in *C. albicans*. Additionally, a more severe filamentation defect (~50%) was observed in *erg251Δ/Δ* compared to the wild-type (Fig 5A and 5B). Complementation of *erg251Δ/Δ* with the *ERG251-A* allele, not the *ERG251-B* allele, was able to partially restore filamentation (Fig 5A and 5B). Taken together, this data supports a dominant role of *ERG251-A* in regulating filamentation.

*ERG251-A* regulation of filamentation might be caused by the control of genes that are involved in the yeast-to-hyphae transition. Transcriptional analysis revealed that deletion of *ERG251-A* (*erg251Δ/ERG251*) resulted in a greater impact on overall gene expression than deletion of *ERG251-B* (*ERG251/erg251Δ*) (Figs 5C and S3A–S3D). In the YPAD condition, deletion of *ERG251-A* resulted in 357 differentially expressed genes (log2 fold change ≥ 0.5 or ≤-0.5 and adjusted *p*-value < 0.1, S5 Table) compared to deletion of *ERG251-B* which altered expression of 26 genes (log2 fold change ≥ 0.5 or ≤-0.5 and adjusted *p*-value < 0.1, S6 Table) (Fig 5D). Only 11 genes were significantly differentially expressed in both heterozygous mutants including *ERG6*, *ERG251* and *CRZ2* (Fig 5D). Notably, *ERG6* had increased expression in both heterozygous mutants which may contribute to the activation of the alternate pathway for ergosterol biosynthesis (Figs 5D, S3A, and S3B). This suggests there is redundancy of the *ERG251-A* and *ERG251-B* alleles in ergosterol biosynthesis, and supports why loss of function of either allele results in the same azole tolerance phenotype (Fig 1). Furthermore, in the SC5314 background *ERG251-A* and *ERG251-B* had similar RNA abundance (S3F Fig), and both of the tagged proteins, Erg251-A-GFP and Erg251-B-GFP, localized to the endoplasmic reticulum (ER) in both yeast and hyphal phases (S3G Fig). This indicates that the divergent function of the two *ERG251* alleles is not caused by allelic expression or subcellular translocation. Among the 346 genes that were differentially expressed only in *erg251Δ/ERG251*, GO analysis revealed an enrichment of genes that regulate filamentation (S3C Fig and S7 Table). Genes that positively regulate filamentation, including *HYR1* and *HWP1*, and their up-stream transcription factor *SFL2* were all down-regulated in *erg251Δ/ERG251* (Figs S3A and 5E) [56,57]. Transcription factor *EFH1* was up-regulated in *erg251Δ/ERG251* and its overexpression may lead to pseudohyphal formation (Figs S3A and 5E) [67,68]. Finally, we found that in YPAD, both *erg251Δ/ERG251* and *erg251Δ/Δ* have largely conserved regulation of this subset of genes involved in filamentation: *SFL2*, *HWP1*, *HWP2*, *HYR1*, *HYR3*, and *EFH1* (Fig 5E).

### Deletion of *ERG251* disrupts the ergosterol production and reduces accumulation of the toxic dienol in the presence of FLC

Homozygous deletion of *ERG251* results in a diverse set of phenotypic effects that may be directly related to disrupted ergosterol biosynthesis and lipid metabolism. To more comprehensively analyze the impact of deleting *ERG251* on ergosterol biosynthesis, we first analyzed transcription of genes involved in ergosterol biosynthesis from all three pathways: mevalonate,

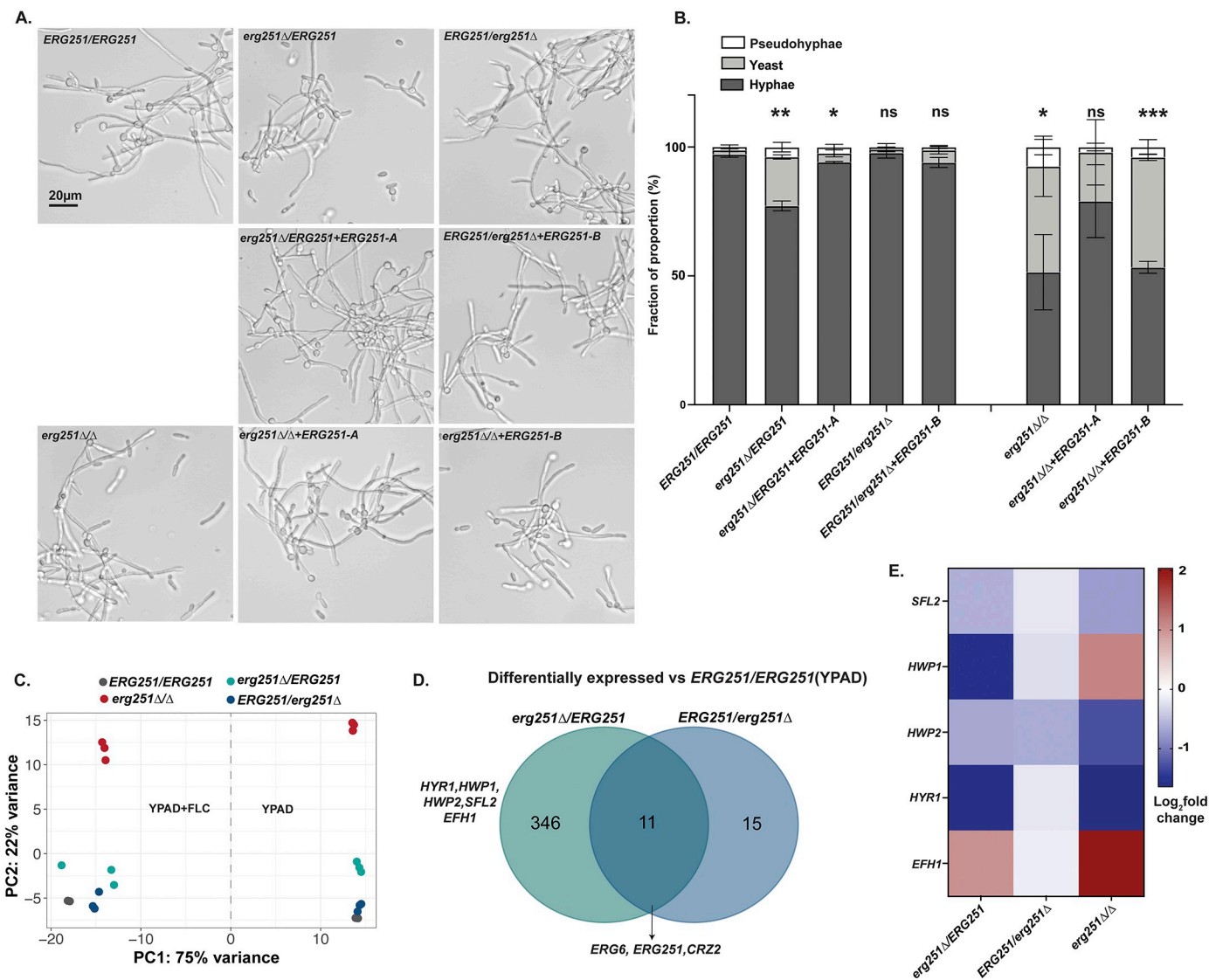

**Fig 5. Deletion of *ERG251-A* but not *ERG251-B* leads to decreased filamentation. A.** Representative filamentation images of wild-type *ERG251/ERG251*, *erg251Δ/ERG251*, *erg251Δ/ERG251+ERG251-A*, *ERG251/erg251Δ*, *ERG251/erg251Δ+ERG251-B*, *erg251Δ/Δ*, *erg251Δ/Δ+ERG251-A*, and *erg251Δ/Δ+ERG251-B*. Cells were induced in RPMI supplemented with 10% FBS for 4 hrs. Scale bar, 20 μm. **B.** Quantification of the yeast (<6μm), pseudohyphae (15–36 μm), and hyphae (>36 μm) from genotypes in Fig 5A. 150 to 500 cells were counted for each strain, and at least two biological replicates were performed. Values are mean ± SEM calculated from three biological replicates. Statistical significance for filamentation was compared to *ERG251/ERG251* and assessed using two-way ANOVA with uncorrected Fisher's LSD, ***$P$ <0.001, **$P$ <0.01, * $P$ ≤ 0.05, ns: $P$ >0.05. **C.** Principal component analysis of transcriptional data in YPAD and YPAD+FLC (1μg/ml) for *ERG251/ERG251*, *erg251Δ/ERG251*, *ERG251/erg251Δ*, and *erg251Δ/Δ*. **D.** Venn diagrams comparing the genes that are differentially expressed in *erg251Δ/ERG251* and *ERG251/erg251Δ* (log2 fold change ≥ 0.5 or ≤-0.5 and adjusted $p$-value < 0.1) relative to *ERG251/ERG251* in YPAD. **E.** The relative expression level (log2 fold change) of genes associated with filamentation in *erg251Δ/ERG251*, *ERG251/erg251Δ*, and *erg251Δ/Δ* compared to *ERG251/ERG251* in YPAD.

late and alternate (Fig 6A). In YPAD, *erg251Δ/Δ* had decreased expression relative to the wild-type of 11 *ERG* genes and increased expression of *ERG12*, *ERG25*, and *ERG6* (Figs 6A and S4A). Among the 11 down-regulated *ERG* genes, *ERG1* and *ERG11* had the most significant decreases (log2 fold change = -1.5 and -1.3 respectively). These two genes represent two rate-limiting steps in the ergosterol biosynthesis pathway [69]. Two additional key genes were

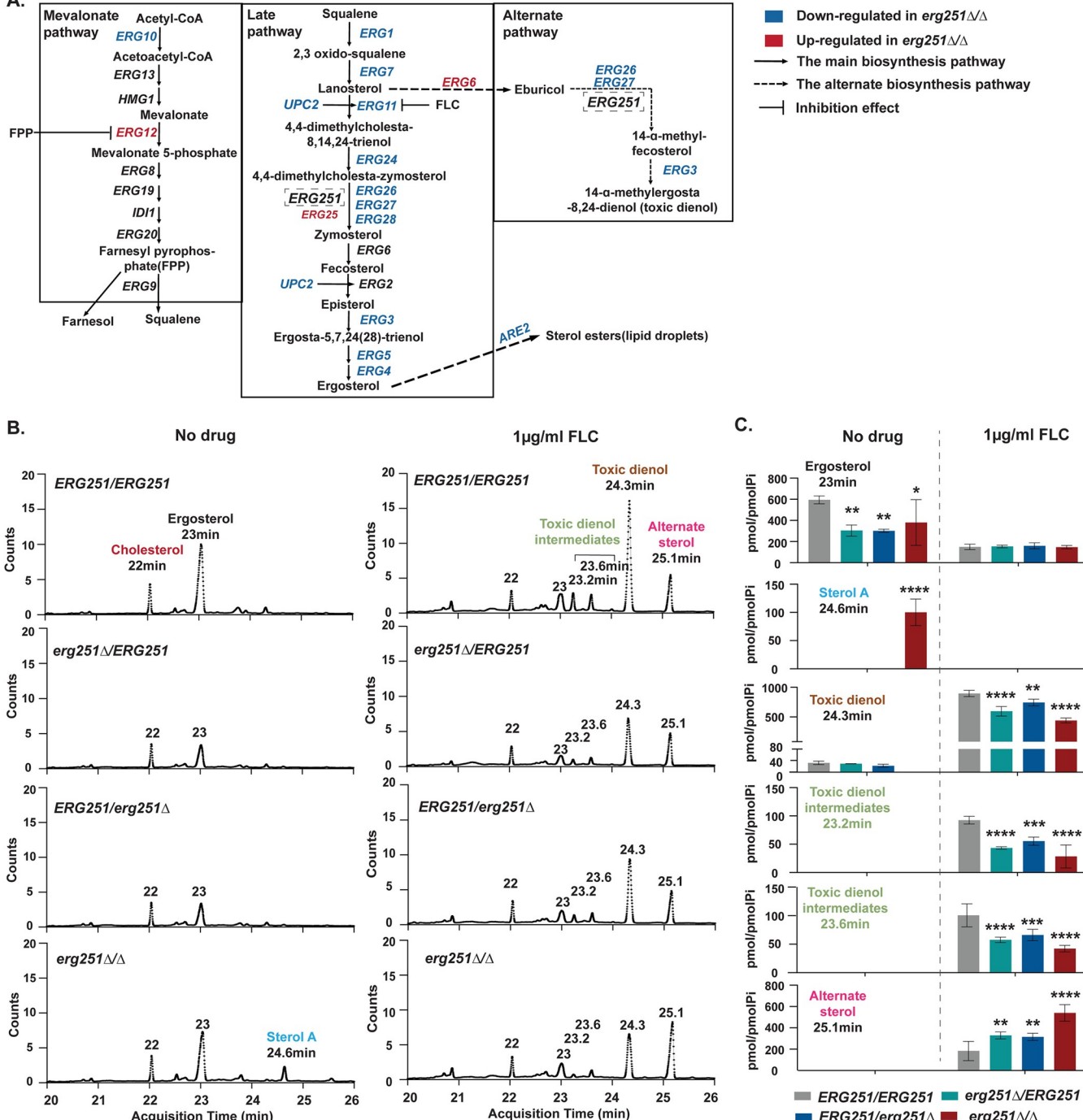

**Fig 6. Homozygous deletion of *ERG251* leads to the decreased ergosterol accumulation in the absence of FLC and decreased production of toxic dienol in the presence of FLC. A.** Overview of the ergosterol biosynthetic pathway in *C. albicans*, including the mevalonate, late ergosterol, and alternate pathways [11,19,69,73]. Genes that were down-regulated (blue) and up-regulated (red) in the *erg251Δ/Δ* under no drug conditions relative to SC5314 (S4A Fig). **B.** Representative GC-MS profiling of *ERG251/ERG251*, *erg251Δ/ERG251*, *ERG251/erg251Δ*, and *erg251Δ/Δ* strains in absence of drug and in the presence of 1μg/ml FLC. The number above each peak represents the area of the peak based on the number of counts taken by the mass spectrometer detector at the point of retention. Labelled peaks indicate the input standard cholesterol, ergosterol, and unidentified sterols: sterol A, toxic dienol intermediates and alternative sterol. All unidentified sterols were compared with known standards: ergosterol, lanosterol, obtusifoliol, zymosterol, 4,4-dimethyl zymosterol, eburicol, episterol, and gramisterol (24-methylenelophenol). **C.** Abundance of sterols in tested strains from Fig 6B. Values are mean ± SEM calculated from three biological replicates. Statistical significance for filamentation was compared to *ERG251/ERG251* from the same condition and assessed using two-way ANOVA followed by Dunnett's multiple comparisons test", ****$P$ <0.0001, ***$p$<0.001, **$p$< 0.01, *$p$<0.05, ns: $P$ >0.05.

down regulated in *erg251Δ/Δ* compared to wild-type in YPAD: *UPC2*, encoding a transcription factor that activates *ERG* genes, and *ARE2*, encoding a sterol acyltransferase that regulates the storage and decomposition of ergosterol [70–72]. When comparing the transcriptional abundance of *ERG* genes between the two growth conditions (FLC vs YPAD), we found that almost all *ERG* genes had increased expression in response to FLC exposure across all four strains, with or without *ERG251* deletion (S4A Fig and S12–S15 Tables). Strikingly, *ERG6* had 8-fold increased expression in *erg251Δ/Δ* in the presence of FLC relative to wild-type (S4B Fig and S11 Table), and over-expression of *ERG6* can result in accumulation of the alternative sterols leading to cell survival in the presence of FLC [19,25,73]. Therefore, we hypothesize that deletion of *ERG251* can disrupt the ergosterol production and lead to accumulation of alternative sterols in the presence of FLC.

To test this hypothesis, we performed gas chromatography-mass spectrometry (GC-MS) analysis of sterol accumulation for the SC5314 wild-type (*ERG251/ERG251)*, two *ERG251* heterozygous deletion mutants, and one homozygous deletion mutant in the absence or presence of 1ug/ml FLC. In the absence of FLC, all *ERG251* deletion mutants exhibited significantly decreased ergosterol accumulation compared to wild-type (Fig 6B and 6C). Notably, in the absence of FLC, the *erg251Δ/Δ* mutant exhibited a unique peak at 24.60 min retention time (Fig 6B), and the mass spectrum of this peak had 63% similarity to 4-methyl episterol (also known as 24-methylenelophenol or gramisterol) (S5 Fig). This indicates that in the absence of FLC, *ERG251* deletion (heterozygous or homozygous) leads to the disruption of ergosterol biosynthesis, but only homozygous deletion of *ERG251* results in accumulation of an ergosterol intermediate that likely contributes to the severe growth defect and membrane sensitivity of *erg251Δ/Δ* (Figs 3A and 4C). We named this 4-methyl episterol-related ergosterol intermediate "sterol A" here because the elution profile matches the unidentified sterol A detected when an *erg251Δ/Δ* mutant was cultured under biofilm growth conditions [74]

In presence of 1ug/ml FLC, all *ERG251* deletion mutants exhibited significantly decreased accumulation of the toxic dienol (peaks at 24.3 min) and its intermediates (peaks at 23.2 and 23.6 min) compared to wild-type (Fig 6B and 6C). Importantly, FLC exposure combined with *ERG251* deletion resulted in increased accumulation of an alternative sterol (retention time at 25.1 min, Fig 6B and 6C) with a similar spectral profile as the lanosterol standard (not the eburicol or 4,4-dimethyl zymosterol standards), possibly the lanosterol derivative 24-methylenedihydrolanosterol (S5 Fig) [75]. This indicates that in the presence of FLC, *ERG251* deletion (heterozygous and homozygous) reduces the accumulation of toxic dienol production and increases the accumulation of non-toxic alternative sterols causing azole drug tolerance.

### *ERG251* is the major active C-4 sterol methyl oxidase in the alternate sterol pathway controlling drug susceptibility

*ERG25*, the paralog of *ERG251*, is expressed at low levels relative to *ERG251* in wild-type cells [18]. We found that *ERG25* expression was increased upon deletion of *ERG251* (both heterozygous and homozygous deletion) in the absence of FLC (log2 fold change ~0.5, S4A Fig). Recently, Xiong et al. reported that increased expression of *ERG25* alone can improve the growth and filamentation defects of an *erg251Δ/Δ* null mutant [74]. Therefore, we hypothesized that Erg25 and Erg251 can compensate for each other during ergosterol biosynthesis despite having significant sequence divergence (S6A Fig). Consistent with this, a double homozygous deletion of *ERG251* and *ERG25* was not possible after multiple attempts. This indicates that there is some essential compensation between the two enzymes, most likely in the late ergosterol biosynthesis pathway. A heterozygous deletion of *ERG25* in either the wild-type or *erg251Δ/Δ* strain backgrounds had no effect on drug susceptibility (S6B and S6C Fig).

Surprisingly, in the absence of FLC, wild-type and heterozygous deletion mutants of *ERG251* generate low amounts of the toxic dienol, but this toxic dienol was completely eliminated in the *erg251Δ/Δ* null mutant (Fig 6C). We conclude that Erg251 is the solo C-4 sterol methyl oxidase in the alternate sterol pathway under typical growth conditions (Fig 6A). However, in the presence of FLC, *ERG25* expression was increased along with *ERG6* in the *erg251Δ/Δ* null mutant (S4A and S6D Figs) and this combination may be sufficient to provide flux through the alternate sterol pathway, resulting in accumulation (albeit significantly reduced) of the toxic dienol and dienol intermediates (Fig 6C).

## Zinc transporter contributes to Erg251-mediated azole tolerance

To determine the mechanism driving decreased drug susceptibility, we further compared the transcriptional analysis of all three *ERG251* deletion mutants during growth in FLC. No significant change in expression was observed for genes encoding the drug efflux pumps *CDR1*, *CDR2*, and *MDR1* across the three *ERG251* deletion mutants, compared to wild-type (log2 fold change $\geq$ 0.5 or $\leq$ -0.5 and adjusted *p*-value < 0.1), with an exception for *MDR1* in *erg251Δ/Δ* that had a 2-fold increase (adjusted *p*-value = $9.6 \times 10^{-7}$) (Fig 7A and S9–S11 Tables). We next determined whether there was a conserved transcriptional response across the three *ERG251* mutants after FLC exposure. In YPAD+1ug/ml FLC, only 8 genes were significantly differentially expressed in all three *ERG251* deletion mutants relative to wild-type (heterozygous deletion mutants: log2 fold change $\geq$ 0.5 or $\leq$ -0.5 and adjusted *p*-value < 0.1, homozygous deletion mutant: log2 fold change $\geq$ 1 or $\leq$ -1 and adjusted p-value < 0.05) (Fig 7B). Based on the predicted and characterized functions of these 8 genes, we focused on *ZRT2* that encodes a zinc transporter that localizes to the plasma membrane and is essential in *C. albicans* for Zinc uptake and growth at acidic pH [76]. *ZRT2* was upregulated ~1.5 fold in both the heterozygous and homozygous *ERG251* deletion mutants during FLC exposure (Fig 7A–7C).

To delineate the role of Zrt2 in Erg251-mediate azole tolerance, we engineered additional strains and quantified drug susceptibility. Overexpression of *ZRT2* in the wild-type SC5314 background resulted in an ~1.6-fold increase in mRNA expression relative to wild-type in the presence of FLC (Fig 7C). Overexpression of *ZRT2* caused increased FLC tolerance (SMG = 0.24–0.25) relative to wild-type, however less tolerance than the *ERG251* heterozygous deletion mutants (SMG = 0.6) (Fig 7D). To test if Zrt2 directly contributes to the high tolerance observed in *ERG251* heterozygous deletion mutants, we deleted a single copy of *ZRT2* from the *erg251Δ/ERG251* background (Fig 7D). Both independent transformants exhibited reduced FLC tolerance (SMG = 0.45–0.49) compared to the *ERG251* heterozygous deletion mutant (SMG = 0.6) (Fig 7D). Taken together, we conclude that Zrt2 directly contributes to *ERG251*-mediated azole tolerance together with sterol composition changes.

## Mutants with single allele dysfunction of *ERG251* maintain pathogenicity in a murine model

We next explored the effects of *ERG251* mutations during infection. The pleiotropic effects of *ERG251* on varied cellular responses, especially decreased resistance to superoxide and reduced filamentation raise the question of whether dysfunction of *ERG251* would also lead to a defect in pathogenicity. We tested the two heterozygous and one homozygous *ERG251* deletion mutants in the standard mouse tail-vein injection model of disseminated candidiasis [77]. There was no difference in survival between the wild-type control SC5314 and the heterozygous mutants (*erg251Δ/ERG251* and *ERG251/erg251Δ*) (Fig 8). However, mice infected with *erg251Δ/Δ* had significantly longer survival compared to the wild-type control (*P* = 0.0015, Long-rank (Mantel-Cox) test) (Fig 8). We also tested the survival of mice infected with the two

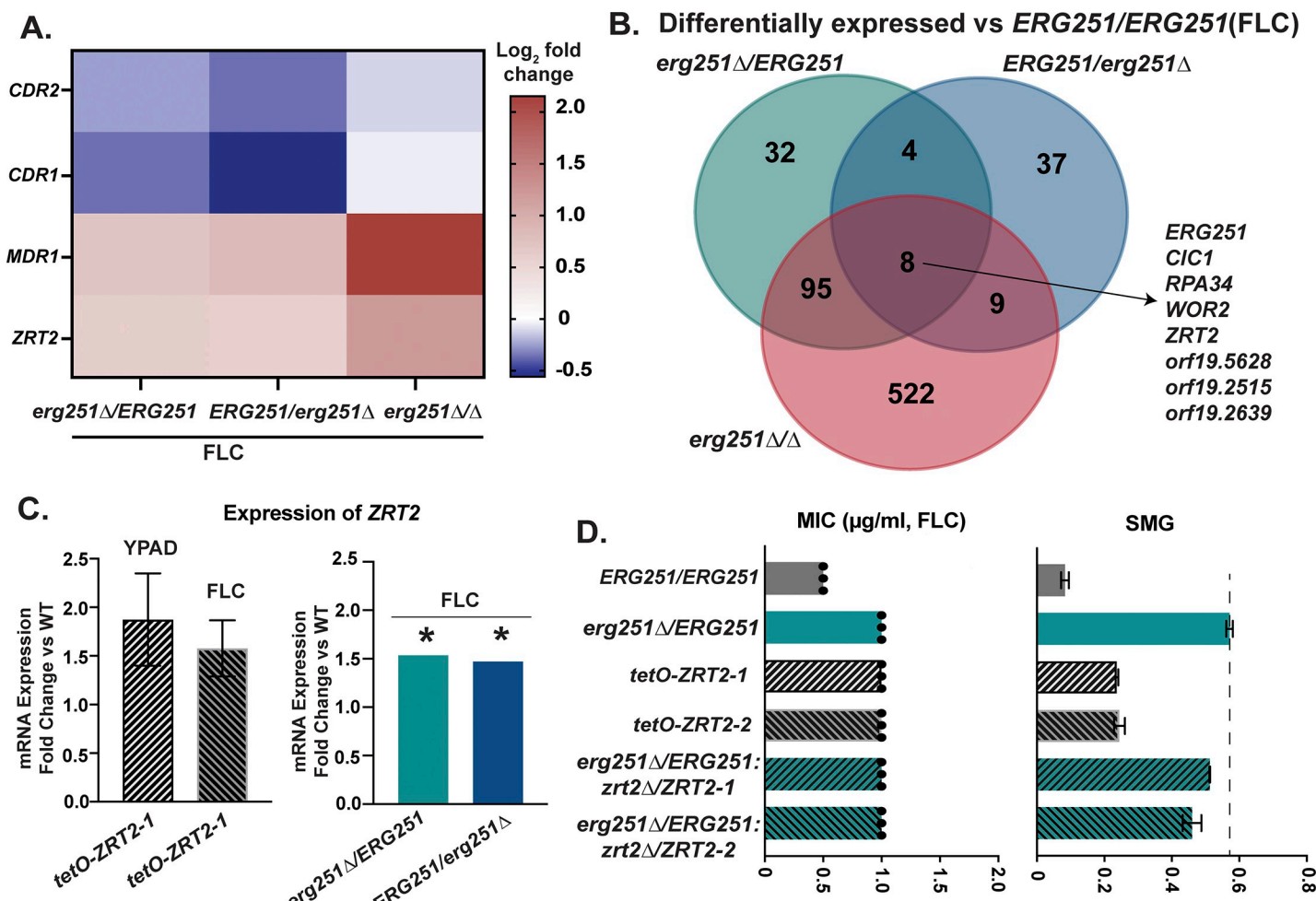

**Fig 7. Dysfunction of *ERG251* activates a Zinc transporter contributing to decreased azole susceptibility. A.** The relative expression level (log2 fold change) of *CDR1, CDR2, MDR1 and ZRT2* in *erg251Δ/ERG251*, *ERG251/erg251Δ*, and *erg251Δ/Δ* compared to *ERG251/ERG251* under YPAD+1μg/ml FLC condition. **B.** Venn diagrams comparing the genes that differentially expressed in *erg251Δ/ERG251*, *ERG251/erg251Δ* and *erg251Δ/Δ* relative to *ERG251/ERG251* under YPAD+1μg/ml FLC condition. **C.** mRNA expression fold change (y-axis) of *ZRT2* in *ZRT2* overexpression strain (*tetO-ZRT2-1*) (Left, RT-qPCR) and in *erg251Δ/ERG251* and *ERG251/erg251Δ* (Right, RNA-seq) under YPAD or YPAD+1μg/ml FLC condition relative to the wild-type control. Asterisk indicates the expression change is significant (adjusted p-value < 0.1). Dotted line indicates the SMG of *erg251Δ/ERG251*. At least three biological repeats were performed. **D.** 24hr MIC (left, μg/ml) and 48hr SMG (right, tolerance) in FLC for two *ZRT2* overexpression strains (*tetO-ZRT2-1* and *tetO-ZRT2-2*, independent transformants) in SC5314 background and two *ZRT2* heterozygous deletion mutants in *erg251Δ/ERG251* background (*erg251Δ/ERG251*: *zrt2Δ/ZRT2-1* and *erg251Δ/ERG251*: *zrt2Δ/ZRT2-2*) together with SC5314 (*ERG251/ERG251*) and *erg251Δ/ERG251* as the controls. MIC: each dot represents a single replicate and bar represents the average of three technical replicates of a single strain; SMG values are mean ± SEM calculated from three technical replicates of a single strain. At least three biological replicates were performed.

complementation strains of *erg251Δ/Δ*, and both complementation strains restored virulence (Fig 8). Taken together, this indicates that mutants with homozygous deletion of *ERG251* have attenuated virulence, which supports the importance of *ERG251* in varied cellular responses essential for pathogenicity. However, the azole tolerant mutants with a single allele dysfunction of *ERG251* remained infectious.

## Discussion

### Changes in *ERG251* impact the susceptibility of *C. albicans* to azoles

Antifungal tolerance varies among *C. albicans* clinical isolates and correlates with the inability to clear an infection. The tolerance phenotype is stable even in the absence of antifungal drug

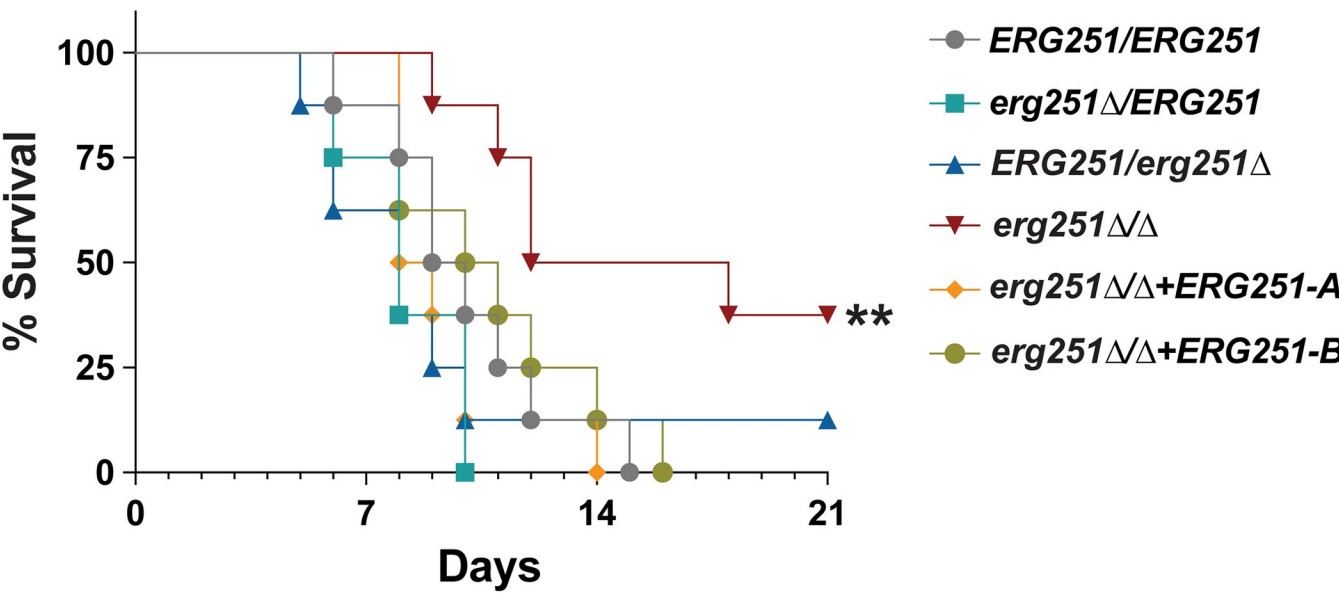

**Fig 8.** ***ERG251*** **heterozygous deletion mutants maintain their pathogenicity in a murine model. A.** ICR mice were injected via the tail vein with 5x10$^5$ cells of *ERG251/ERG251* (SC5314), *erg251Δ/ERG251*, *ERG251/erg251Δ*, and *erg251Δ/Δ+ERG251-A* and *erg251Δ/Δ+ERG251-B* and survival was presented over the time. The *erg251Δ/Δ* mutant survival curves were significantly attenuated from that of the *ERG251/ERG251* (Log-rank (Mantel-Cox) test; **, *p* = 0.0015). Eight mice per strain were used.

stress [9]. Despite this, the molecular mechanisms causing antifungal tolerance are not known. We found that *ERG251* is a hotspot for point mutations during adaptation to antifungal drug stress, and that heterozygous deletion of *ERG251* can drive azole tolerance in diverse clinical isolates of *C. albicans*. This is the first example of loss-of-function point mutations that cause azole tolerance in *C. albicans*. The mechanism of *ERG251*-mediated azole tolerance is caused by decreased accumulation of the toxic dienol and increased non-toxic alternative sterols as well as increased expression of the zinc transporter *ZRT2*.

Aneuploidy is frequently associated with the rapid acquisition of azole tolerance [78,79]. We recently found that the Chr3 and Chr6 concurrent aneuploidies conferred multi-azole tolerance via elevated drug efflux [Zhou et al. 2024, accepted]. Here, we identified recurrent, heterozygous loss-of-function mutations in *ERG251* (Chr4) that occurred together with aneuploidies of smaller chromosomes (Chr3-Chr7) during adaptation to FLC (Table 1). The combination of aneuploidy and *ERG251* dysfunction, two mechanisms that independently cause azole tolerance, resulted in *bona fide* azole resistance. Multiple simultaneous mutations are common in cancer cells and often lead to treatment failure [80–83]. Our results highlight the diverse trajectories that *C. albicans* can take during adaptation to antifungal drugs and support that two independent mechanisms of tolerance (point mutation and aneuploidy) can evolve within the same cell resulting in drug resistance.

## Drug susceptibility in heterozygous versus homozygous *ERG251* deletion mutants

One of the particularly striking observations from our study was the range of different heterozygous point mutations that phenocopied the heterozygous deletion of *ERG251*. The numerous possible mutation sites at either allele of the *ERG251* combined with a strong fitness advantage in FLC for all characterized *ERG251* mutations explains why *ERG251* mutations

were recurrent in three independent *in vitro* evolution experiments. Point mutations in ergosterol-related genes like *ERG11* and *UPC2* that cause drug resistance in *Candida* species are frequently homozygous in diploid organisms and result in higher MIC than heterozygous mutations [84–89]. Comparatively, all of the *de novo ERG251* point mutations identified here were heterozygous. Although the homozygous deletion of *ERG251* led to a similar fitness advantage as the heterozygous deletion mutants in the presence of low concentrations of FLC (<1µg/ml), the homozygous deletion strain exhibited a fitness cost in rich medium and at higher concentrations of FLC, supporting why only heterozygous mutants were identified during the *in vitro* evolution experiments.

Prior studies of the relationship of *ERG251* to azole susceptibility in *C. albicans* have found conflicting results. In some experiments, loss of *ERG251* through genetic manipulation or pharmacological inhibition resulted in increased susceptibility to azoles whereas in other experiments, disruption of *ERG251* decreased susceptibility to azoles [18,34]. Our findings help to explain these disparate results. During exposure to low concentrations of FLC, both heterozygous and homozygous *ERG251* deletion mutants had sterol composition changes that support their acquired azole tolerance: decreased accumulation of the toxic dienol and increased accumulation of alternative non-toxic sterol intermediates. However, we found *erg251Δ/Δ* had decreased cell growth in high concentrations of FLC. This is in contrast to the slow growth of two heterozygous deletion mutants despite the FLC concentration (drug tolerance). Therefore, depending on the concentration of FLC, loss of *ERG251* can increase or decrease growth of *C. albicans*. Lu *et al.* detected much more disrupted ergosterol production in the *erg251Δ/Δ* mutant in the presence of high concentration of FLC [18]. This indicates that the high concentration of FLC might pose much more severe inhibition on the ergosterol accumulation in *erg251Δ/Δ* reducing the cell growth and resulting increased drug susceptibility.

## The impact of *ERG251* homozygous deletion on metabolic process of farnesol

In the absence of FLC, *ERG251* heterozygous and homozygous deletion mutants all had decreased accumulation of ergosterol, however only the *ERG251* homozygous deletion mutant exhibited accumulation of 4-methyl episterol ("sterol A") which may directly contribute to the growth defect of and severe membrane sensitivity of *erg251Δ/Δ*. However, the growth defect is not likely to be due only to sterol A as we also observed a connection between farnesol and ergosterol biosynthesis. We found moderate concentrations of farnesol (62.5–250 µM) improved the growth of *erg251Δ/Δ* in rich media, while farnesol had no impact on wild-type growth. Therefore, homozygous deletion of *ERG251* may result in disrupted ergosterol biosynthesis which subsequently provides negative feedback on farnesol production contributing to the growth defect of *erg251Δ/Δ*. One *ERG* gene, *ERG12*, had decreased expression in response to FLC—the opposite trend from all other *ERG* genes (Fig 6). *ERG12* encodes mevalonate kinase and converts mevalonate into 5-phosphomevalonate, the precursor for farnesyl pyrophosphate (FPP). We hypothesize that the FLC-induced repression of *ERG12* is caused by negative feedback resulting from an increase in farnesol or FPP that occurs upon exposure to FLC. Homozygous deletion of *ERG251* correlated with increased expression of *ERG12*. The increased expression of *ERG12* combined with improved growth of *erg251Δ/Δ* mutant when supplied with moderate farnesol, suggests that *ERG12* expression is negatively regulated by farnesol or its precursor, FPP, and that both deletion of *ERG251* and FLC exposure impact *ERG12* expression possibly via farnesol production (S6E Fig) [90]. Importantly, we found farnesol can make FLC fungicidal at high concentrations. The inhibition of high concentration

farnesol on *ERG12* might further pose a block on the mevalonate pathway which together with disrupted ergosterol production from FLC results in a killing effect on fungal cells. This also provides evidence that dysfunction of Erg251 can also pose impacts on other metabolic processes affecting different phenotypes.

## Changes in sterol composition resulting from *ERG251* inactivation have pleiotropic effects

Homozygous deletion of *ERG251* had a global impact on the expression of sterol biosynthetic genes resulting in down-regulation of ergosterol biosynthetic genes and pleiotropic effects of genes encoding cell surface proteins. The different regulation of these genes had direct correlation with different phenotypes observed for *erg251Δ/Δ*. Furthermore, our analysis of sterol composition for *ERG251* mutants supports that *ERG251* deletion can cause altered membrane composition, even in the absence of FLC. Previous studies found that the GPI biosynthesis of cell surface proteins can affect ergosterol level via regulating *ERG11* [66]. Abnormal ergosterol production and accumulation of ergosterol intermediates can lead to transcriptional activation of stress responses [12,91]. Therefore, we propose that the altered membrane composition of *ERG251* homozygous mutant causes pleiotropic effects on global gene expression and localization of cell surface proteins that result in the observed phenotypic changes

Furthermore, our sterol data also provide evidence for the divergent roles and substrate preferences of *ERG251* and *ERG25* in ergosterol and alternate sterol pathways. The shared enzymatic function and regulatory networks of paralogs supports a model where compensation between *ERG251* and *ERG25* occurs at the level of gene expression [39,92]. We and Xiong *et al.* identified the sustained production of ergosterol in *erg251Δ/Δ* mutants without accumulation of 4,4-methyl zymosterol, the common substrate for C-4 methyl sterol oxidases [74]. These data indicate that there is compensation between the paralogs *ERG251* and *ERG25* during ergosterol biosynthesis under normal growth conditions and that substrate preference is similar during ergosterol biosynthesis. The same ergosterol intermediate ("sterol A") was detected only in *erg251Δ/Δ* in both studies, and we characterized it as 4-methyl-episterol. This suggests that *ERG251* has a substrate preference for 4-methyl-episterol during ergosterol production. We propose that this is similar to *Chlorella sorokiniana*, where enzymes in the ergosterol biosynthesis pathway can have multiple possible substrates to support flux through ergosterol production upon azole inhibition [93].

No toxic dienol was produced in the *erg251Δ/Δ* mutant in the absence of FLC. This supports our model that *ERG251* is the solo C-4 sterol methyl oxidase in the alternate sterol pathway under typical growth conditions. With FLC exposure, partial dysfunction of *ERG251* is sufficient to alter sterol composition with accumulation of an alternative sterol that is predicted to be 24-methylenedihydrolanosterol. This alternative sterol is different from eburicol, another common substrate for C-4 sterol methyl oxidase, that Lu *et al.* predicted to accumulate in the *ERG251* null mutant in the presence of high levels of fluconazole [18]. Importantly, we included an eburicol standard in our GC-MS analysis and do not detect eburicol in our strains. This suggests that there might be different possible substrates for *ERG251* in the alternative pathway and that the degree in which the ergosterol biosynthesis pathway is inhibited by different azole concentrations might impact the substrate preference. The sterol changes of *ERG251* deletion mutants also caused decreased drug susceptibility, while deletion of *ERG25* in wild-type or *erg251Δ/Δ* strains had no impact on drug susceptibility. In conclusion, in *C. albicans* Erg251 functions as the major active C-4 sterol methyl oxidase controlling drug susceptibility, filamentation, biofilm formation and other stress responses [18,74].

In summary, this study identified recurrent heterozygous point mutations in the methyl sterol oxidase *ERG251* during adaptation to antifungal drug stress and characterized the first example of point mutation-driven azole tolerance. We utilized genetic, transcriptional and phenotypic analyses to understand the effects of inactivating *ERG251* both partially and completely. Increased azole tolerance was observed in two distinct genetic backgrounds, and heterozygous loss-of-function mutations of *ERG251* promote multi-azole tolerance while maintaining virulence in a mouse model of systemic infection. This suggests that as the main C-4 sterol methyl oxidase, Erg251 is not an ideal drug target as proposed previously [18,40], as it is a hotspot for mutation-driven azole tolerance and mutants can sustain pathogenicity.

## Materials and methods

### Ethics statement

The mouse experiments were approved by the Institutional Animal Care and Use Committee of the Lundquist Institute for Biomedical Innovation at Harbor-University of California, Los Angeles Medical Center.

### Yeast isolates and culture conditions

All strains used in this study are listed in S1 Table including FLC evolved isolates and engineered yeast and bacteria strains. Strains were stored at -80˚C in 20% glycerol. Isolates were grown in YPAD media (20 g/L peptone, 10 g/L yeast extract, 2% dextrose, and 15 g/L agar for plates) supplemented with 40 µg/ml adenine and 80 µg/ml uridine. Cells used for lipid assay were grown in CSM media (6.7g/L Difco yeast nitrogen base without amino acids and with ammonium sulfate, 2g complete amino acid mix, 2% dextrose).

### Strain construction

All engineered strains in this study were generated in the SC5314 background, except one *ERG251* heterozygous deletion in the P75063 background. Strains were constructed by lithium acetate transformation using PCR products with at least 140 bp of homology to the target locus. Primers used in this study are listed in S16 Table.

i. *ERG251* heterozygous deletion
   The *FLIP-NAT* construct was PCR amplified from the plasmid pJK863 [94] using primer sets 1630+1631 and transformed into background strains SC5314 and P75063. NAT-resistant transformants were PCR screened for correct integration of the *FLIP-NAT* construct at the *ERG251* locus using primer pairs 1652+1045 (left of integration) and 1636+1653 (right of integration). Transformants were validated by whole genome sequencing for correct integration.

ii. *ERG251* homozygous deletion
   To promote FLIP-mediated excision of *FLIP-NAT*, correct heterozygous deletion strains *erg251Δ/ERG251* and *ERG251/erg251Δ* were inoculated in YNB+BSA from frozen stocks and incubated at 30˚C, 220rpm, for 48 hrs. Cultures were diluted and 100 cells were plated on YPAD agar, then incubated at 30˚C for 24 hrs. Recovered colonies were patched to both YPAD and YPAD+150 µg/ml NAT. Colonies growing on only YPAD were screened for correct FLIP-mediated excision of *FLIP-NAT* using primer pairs 1574+1575 (inside NAT) and 1652+1653 (across *ERG251*). Colonies that correctly excised *FLIP-NAT* were re-transformed with the *FLIP-NAT* construct (PCR amplified from the plasmid pJK863 [94] using primer sets 1630+1631). NAT-resistant transformants were PCR screened for correct

integration of the *FLIP-NAT* construct at the remaining *ERG251* locus using primer pairs 1652+1045 (left of integration), 1636+1653 (right of integration), and 1632+1633 (inside *ERG251*). Transformants were validated by whole genome sequencing for correct integration.

iii. Construct *ERG251-NAT* plasmid

To generate *ERG251* mutant complementation and point mutation, we built up an *ERG251-NAT* plasmid by fusing *ERG251* upstream plus gene (1644+1645), *NAT* (1574+1575), and *ERG251* downstream (1646+1647) into the pUC19 backbone (1578+1579). PCR amplified fragments were aligned using NEBuilder HiFi DNA Assembly Cloning Kit following the manufacturer's instructions and transferred into *E. coli*. Ampicillin-resistant transformants were screened using primer pairs 1352+1353 and saved in frozen stocks as pAS3118.

iv. *ERG251* mutant complementation

The wild-type *ERG251* upstream region and genes (A or B) were PCR amplified from heterozygous deletion strains using primer pair 1652+1645. The *NAT* gene and downstream *ERG251* region were PCR amplified from pAS3118 using 1574+1653 primers. SOEing PCR was performed using primer pair 1652+1653. The subsequent *ERG251-NAT* construct was transformed into the *erg251Δ/ERG251*, *ERG251/erg251Δ*, and *erg251Δ/Δ* mutants that had previously excised *FLIP-NAT* as described above. NAT-resistant transformants were PCR screened for correct integration of the *ERG251-NAT* construct using primer pairs 1634+1154 (left integration), 1636+1635 (right integration), and 1634+1635 (across integration). Transformants were validated by whole genome sequencing for correct integration.

v. *ERG251* point mutation

Site-directed mutagenesis using double-primer PCR was used to generate *ERG251* point mutation construct. Primers with the desired mutations were paired with *ERG251-NAT* upstream (1652) or downstream (1653) primer to amplify mutated *ERG251-NAT* construct from pAS3118. Four different point mutations were engineered in this study: L113*(1652 +1649/1648+1653), W265G (1652+1655/1654+1653), E273*(1652+1651/1650+1653), and *322Y (1652+1657/1656+1653). The amplified two fragments were fused using SOEing PCR and transformed into the SC5314 background. Transformants were first PCR screened using primers 1652+1575 (left integration) and 1636+1653 (right integration) for correct integration and then validated by whole genome sequencing for base substitution and mutated allele.

vi. *ERG251* overexpression

The TetO promoter replacement construct was PCR amplified using primer pair 1679 +1680 from plasmid pLC605 [95] and transformed into the SC5314 background strain. NAT-resistant transformants were PCR screened for correct integration of the TetO promoter replacement using primer pairs 1652+1176 (left integration) and 1177+1633 (right integration).

vii. *ERG251-GFP*

The C-terminal *GFP-NAT* construct was PCR amplified from plasmid pMG2120 [96] using the 1925+1926 primer pairs and transformed into the SC5314 background strain. NAT-resistant colonies were PCR screened for correct integration of the *GFP-NAT* construct at the C-terminal end of the *ERG251* locus using primer pairs 1632+1927 (left integration) and 1636+1653 (right integration). Transformants were validated by Sanger sequencing for tagged alleles.

viii. *ERG25* heterozygous deletion

The *FLIP-NAT* construct was PCR amplified from plasmid pJK863 using primer pairs 1921+1922 and transformed into both the SC5314 background strain and *erg251Δ/Δ* mutant strain that had previously excised *FLIP-NAT* as described above. NAT-resistant transformants were PCR screened for correct integration of *FLIP-NAT* at the *ERG25* locus using primer pairs 1923+1045 (left integration) and 1636+1924 (right integration).

ix. *ZRT2* overexpression

The TetO promoter replacement construct was PCR amplified from plasmid pLC605 [95] using primer pair 1932+1933 and transformed into the SC5314 background strain. NAT-resistant transformants were PCR screened for correct integration of the TetO promoter replacement using primer pairs 1934+1176 (left integration) and 1177+1633 (right integration).

x. *ZRT2* heterozygous deletion

The *FLIP-NAT* construct was PCR amplified from plasmid pJK863 using primer pairs 1932 +2002 and transformed into the heterozygous *erg251Δ/ERG251* and *ERG251/erg251Δ* mutants that had previously excised *FLIP-NAT* as described above. NAT-resistant transformants were PCR screened for correct integration of *FLIP-NAT* at the *ZRT2* locus using primer pairs 1934+1045 (left integration) and 2003+1636 (right integration).

## Filamentation

Strains were inoculated in 2% dextrose YPAD from frozen stocks and incubated at 30°C, 220 rpm for 16 hrs. Strains were diluted 1:100 into RPMI+10% FBS, then incubated at 37°C for 4 hrs. Cells were harvested, washed once with PBS, and resuspended in PBS before microscopy. Images were captured using an Olympus IX83 microscope and analyzed using ImageJ v1.54d.

## Microscopy

Erg251-GFP tagged strains were struck on YPAD agar plates from frozen stocks and incubated at 30°C for 24 hrs. Cultures were inoculated in 2% YPAD and incubated at 30°C, 220 rpm for 16 hrs. Cultures were diluted 1:100 in fresh 2% YPAD or RPMI+10% FBS, then incubated at 30°C, 220 rpm for 4 hrs. Cells grown in RPMI+10%FBS were spun down, washed once with PBS, and resuspended in PBS before microscopy. Cells grown in YPAD were spun down, washed once with PBS, and resuspended in HBSS+1μg/ml BODIPY ER tracker (Fisher Scientific). Cells were incubated 30 minutes at 37°C, washed twice with HBSS, then resuspended in HBSS before microscopy. Images were captured using an Olympus IX83 microscope.

## Spot plate assay

Strains were inoculated in 2% YPAD from glycerol stocks and incubated at 30°C, 220 rpm for 16 hrs. Cultures were normalized to $10^6$ cells/ml, then 10-fold serially diluted. 10μl of each ($10^6$–$10^3$) dilutions were spotted onto YPAD agar with and without drugs. All spot plates were performed in triplicates. Plates were incubated for 48 hrs at 30°C and imaged using a BioRad GelDoc XR+ imaging system.

## RNA sequencing

i. RNA extraction: For RNA extraction, all 4 strains (wild-type and three *ERG251* deletion mutants) were struck on YPAD agar plates from frozen stocks and incubated at 30°C for 24

hrs. Cultures were then inoculated in 2% YPAD (50 ml) and incubated at 30°C, 220 rpm for 16 hrs. Overnight cultures were then diluted 1:100 into 50 ml YPAD or YPAD+1μg/ml FLC and grown at 30°C, 220 rpm for 5–6 hrs to an $OD_{600}$ of 0.5. Cells were harvested by centrifugation and frozen in liquid nitrogen. RNA were prepared according to the manufacturer's instructions for the Qiagen RNeasy Mini kit (Qiagen, US) using the mechanical disruption method. Removal of DNA was performed with a DNase (Qiagen RNase-free DNase set, US) 1 hr incubation at room temperature on column. Three independent cultures of each strain were grown to provide three biological replicates for RNA-seq experiments.

ii. RNA-Seq: Library preparation was performed by SeqCenter (Pittsburgh, PA) using Illumina's Stranded mRNA preparation and 10bp unique dual indices (UDI). Sequencing was done on a NovaSeq X Plus, producing 150bp paired end reads. Demultiplexing, quality control, and adapter trimming was performed with bcl-convert (v4.1.5) (BCL Convert).

iii. RNA-Seq data analysis: *C. albicans* transcriptome (SC5314_version_A21-s02-m09-r10_orf_coding, downloaded from http://www.candidagenome.org/download/sequence/C_albicans_SC5314/Assembly21/current/?C=S;O=A on 2023/08/17) was indexed using salmon (v1.10.2) [97]. All samples were quasi-mapped to transcriptome index using salmon resulting in quantification of reads mapped to each transcript. The output quantification files were imported into R (v4.1.2) using tximport (v1.22.0) [98] and DESeq2 (1.34.0) [99] was used to model gene expression. PCA analysis was performed using DESeq2 and used to identify any outliers amongst the replicates. We identified one wild-type control grown in FLC as an outlier and excluded this sample from all further analyses (Fig 5C). The DESeq2 'contrast' wrapper was then used to estimate $\log_2$ fold changes for each mutant relative to the wild-type control in YPAD and YPAD+1μg/ml FLC conditions and identify differentially expressed genes. We also estimated $\log_2$ fold changes for each strain grown in FLC relative to the same strain in YPAD and identified differentially expressed genes (S2, S5, and S6 Tables). The threshold for differentially expressed genes was an absolute value $\log_2$ fold change $\geq 0.5$ and adjusted p-value $< 0.1$ for heterozygous deletion mutants. Because the homozygous deletion mutant is predicted to have stronger effects on global gene expression, we used stricter thresholds for the homozygous deletion mutant of an absolute value $\log_2$ fold change $\geq 1$ and adjusted p-value $< 0.05$. Differentially expressed genes in *ERG251* mutants or after FLC exposure are listed in S9–S11 Tables.

## Gene Ontology Analysis

GO slim mapper from *Candida* Genome Database (http://www.candidagenome.org/) [100] was conducted on the set of genes that were differentially expressed in *ERG251* mutants grown in YPAD relative to wild-type controls in YPAD. Process Ontology was performed for all three *ERG251* deletion mutants and output files are included in S3 and S7–S8 Tables.

## Reverse Transcriptase qPCR

i. RNA extraction: For RNA extraction, wild-type (SC5314) and *tetO-ZRT2-1* were struck on YPAD agar plates from frozen stocks and incubated at 30°C for 24 hrs. Cultures were then inoculated in 2% YPAD (50 ml) and incubated at 30°C, 220 rpm for 16 hrs. Overnight cultures were then diluted 1:100 into 50 ml YPAD or YPAD+1μg/ml FLC and grown at 30°C, 220 rpm for 5–6 hrs to an $OD_{600}$ of 0.5. Cells were harvested by centrifugation and frozen in liquid nitrogen. RNA were prepared according to the manufacturer's instructions for the

Qiagen RNeasy Mini kit (Qiagen, US) using the mechanical disruption method. Removal of DNA was performed with a DNase (Qiagen RNase-free DNase set, US) 1 hr incubation at room temperature on column.

ii. RT-PCR: cDNA was prepared using the SuperScript II Reverse Transcriptase (Fisher Scientific) according to the manufacturer's instructions with oligo dT primers and 100 ng of RNA. cDNA was then diluted 1:10 with nuclease-free water for qPCR measurement. Real-time qPCR was conducted using the PowerUp SYBR Green Master Mix (Applied Biosystems) according to the manufacturer's instructions to measure cDNA. Using CFX Connect Real-Time PCR Detection System and Bio-Rad CFX Maestro software to determine Cq values, expression was calculated as the amount of cDNA from the gene of interest relative to the amount of *TEF1* cDNA in the same sample. All primers used in this study are listed in S16 Table.

### Rhodamine 6G efflux assay

Drug efflux was measured using an adapted protocol [27,46]. Strains were struck on 2% YPAD agar from frozen stocks and incubated at 30°C for 24 hrs. Recovered cells were inoculated into 2% YPAD or YPAD+1 μg/ml FLC. Cultures were incubated at 30°C, 220 rpm, for 16 hrs. Cultures were diluted 1:100 into fresh media of the same condition, then incubated 30°C, 220 rpm, for 3 hrs. Subcultures were harvested and washed once with room temperature PBS, then resuspended in PBS and incubated at 30°C for 1 hr. Rhodamine 6G (Sigma) was added to a final concentration of 10 μg/ml. Cells were incubated at 30°C for 1 hr. Following incubation, cells were washed twice with 4°C PBS, then resuspended in room temperature PBS. Immediately, $OD_{600}$ and baseline fluorescence were measured (excitation 344 nm, emission 555 nm) for 5 minutes in 1-minute intervals using a BioTek Synergy H1 plate reader. Following initial measurements, dextrose was added to a final concentration of 1%. Fluorescence was measured for 90 minutes in 2-minute intervals using a BioTek Synergy H1 plate reader. All strains were conducted in three independent replicates and tested with and without dextrose.

### Growth curve assay

Strains were inoculated in 2% dextrose YPAD from frozen stocks and incubated at 30°C, 220 rpm for 16 hrs. Cultures were diluted in fresh 1% dextrose YPAD to a final $OD_{600}$ of 0.01. Normalized cultures were diluted 1:10 into a 96-well NUNC plate containing 1% dextrose YPAD with or without drug. Cells were incubated at 30°C in a BioTek Epoch 2 microplate spectrophotometer shaking in a double orbital (237rpm) with $OD_{600}$ readings taken every 15 minutes for 48 hrs. Each isolate was conducted in triplicates.

### Growth assay for FLC and farnesol

Strains were inoculated in 2% dextrose YPAD from frozen stocks and incubated at 30°C, 220 rpm for 16 hrs. Cultures were diluted in fresh 1% dextrose YPAD to a final $OD_{600}$ of 0.01. Normalized cultures were diluted 1:10 into a 96-well NUNC plate containing 1% dextrose YPAD supplemented with or without drug. Both FLC and farnesol were diluted in a two-fold serial: FLC concentration (x-axis, 2x dilution for each column) ranged from 0 to 256μg/ml, and farnesol concentration (y-axis, 2x dilution for each row) ranged from 0 to 1000μM. Cells were incubated at 30°C in a BioTek Epoch 2 microplate spectrophotometer shaking in a double orbital (237rpm) with $OD_{600}$ readings taken every 15 minutes for 48 hrs. Plates were

conducted in triplicates. 48 hrs later, 10 µl cells from each well were plated onto YPAD agar plate to monitor viability, and plate images were taken after 24 hrs incubation at 30˚C.

### Relative fitness assay

Isolates were inoculated in 2% dextrose YPAD from frozen stocks and incubated at 30˚C, 220 rpm, for 16hrs. Cultures were diluted in fresh 1% dextrose YPAD to a final $OD_{600}$ of 0.01. Normalized cultures from the sample of interest and the fluorescent control strain (same fitness as the WT) were then combined at a 1:1 ratio, and the combined culture was diluted 1:10 into a 96-well NUNC plate containing 1% dextrose YPAD or YPAD+1 µg/ml FLC (initial $OD_{600}$, $N_0$ = 0.001). Cells were incubated at 30˚C in a BioTek Epoch 2 microplate spectrophotometer with double-orbital (237rpm) shaking. OD600 readings were taken every 15 minutes for 48 hrs to monitor cell growth and $OD_{600}$ at the endpoint. 20 µl culture was removed from one of the triplicates for flow cytometry at 24 hrs. Cultures were diluted in PBS and 10,000 singlets were gated and analyzed at each time point using a Cytek Aurora flow cytometer (R0021). After 24 hrs the population reached the stationary phase and $OD_{600}$ was about 1.3 (Nt), therefore there were a total of 10 generations for the competition assay estimated using equation generations = [log10 (Nt/N0)]/0.3. Proportions of sample interest were indicated by the proportion of non-fluorescent cells while fluorescent control was indicated by blue fluorescent cells. All competitive assays were conducted in three independent replicates. Relative fitness was estimated using natural log regression analysis of the proportion of sample of interest and fluorescent control against the generations (10 generations): ln (proportion of sample of interest/ proportion of fluorescent control)/generations.

### Microdilution MIC and SMG assays

Isolates were inoculated in 2% dextrose YPAD from frozen stocks and incubated at 30˚C, 220 rpm, for 16 hrs. Cultures were diluted in fresh 1% dextrose YPAD to a final $OD_{600}$ of 0.01. Normalized cultures were diluted 1:10 into 1% dextrose YPAD media containing either a two-fold serial dilution of drug or a no-drug control. Drug concentrations ranged from 0.5µg/ml to 256 µg/ml FLC and 0.0625 µg/ml to 32 µg/ml itraconazole and voriconazole. Triplicates of each isolate were set up using flat-bottom 96-well plates and incubated in a humidified chamber at 30˚C. Cells were resuspended at the 24 hrs and 48 hrs time points and $OD_{600}$ readings were taken using a BioTek Epoch 2 microplate spectrophotometer. The $MIC_{50}$ of each strain was determined as the drug concentration at which $\geq$ 50% of growth was inhibited relative to the no-drug control at 24 hrs post-inoculation. The supra-MIC growth (SMG) was measured as the average growth above the $MIC_{50}$ when standardized to the no-drug control at 48 hrs post-inoculation [9]. To measure the impact of Hsp90 inhibition, 2.5 µM radicicol (Cayman Chemicals) was added to the 1% dextrose YPAD in the microdilution MIC and SMG assay plate. To determine cell viability, 5µl was removed from the assay plate after the 48 hr time point and plated onto YPAD agar plates without any drugs. Plate images were taken after 24 hrs incubation at 30˚C.

### Ploidy analysis (DNA-PI staining)

Cells were prepared as described previously [101]. Strains were inoculated in 2% dextrose YPAD from frozen stocks and incubated at 30˚C, 220 rpm, for 16 hrs (cell density ~$1x10^7$ cells/ml). Cultures were spun down and the supernatant was removed. Cell pellets were resuspended in 70% ethanol, and then washed twice with 50 mM sodium citrate. Cells were then treated with RNAse A at 37˚C for at least 2 hrs, and then stained with 25 µg/ml propidium iodide (PI) at 37˚C in the dark for 16 hrs. Samples were diluted in 50 mM sodium citrate and

at least 10,000 singlets were gated and analyzed using a Cytek Aurora flow cytometer (R0021). 488-nm lasers were used to excite the PI-staining and 618/24 filters were used to detect the PI-staining emission signals. Data were analyzed using FlowJo v10.8.1.

## Lipid assay

i. For lipid assay, all 4 strains (wild-type and 3 *ERG251* deletion mutants) were struck on YPAD agar plates from frozen stocks and incubated at 30˚C for 24 hrs. Cultures were then inoculated in 2% YPAD (50 ml) and incubated at 30˚C, 220 rpm for 16 hrs. Overnight cultures were collected by centrifugation and then washed once with PBS. After resuspending with 50 ml 2% dextrose CSM media, cells were added into fresh CSM media with 1μg/ml FLC or without drug in 1:50 dilution for another 5–6 hours. Then cell density of the cell suspension was determined to calculate the amount needed ($5x10^8$ cells). Cells were then harvested by centrifugation and frozen in -80˚C.

ii. Lipid extraction and GC-MS was conducted as previously described [102]. Pellets with $5x10^8$ cells were used for lipid extraction. The dried total samples were resuspended in 100 μL chloroform added to 100 μL of BSTFA reagent (Thermo Scientific) and incubated at 70˚C for 1 hour prior to GC-MS (Agilent 7890B GC–MS, Agilent 5977A MSD) analysis [103].

The retention time and mass spectral patterns of a sterol standard were used as references for lipid analysis. Sterol standards used in this study include ergosterol (Smolecule, catalog # S527372), lanosterol (Smolecule, catalog #S532452), obtusifoliol (Smolecule, catalog #S563624), zymosterol (Smolecule, catalog # S580329), 4,4-dimethyl zymosterol (Avanti catalog # 700073), eburicol (Smolecule, catalog # S633611), episterol (Smolecule, catalog # S628882), and gramisterol (24-methylenelophenol or 4-methyl episterol) (Smolecule, catalog # S626191). Cholesterol (Avanti catalog # 700100) was added as an internal standard for these analyses prior to lipid extraction. The relative amount of unknown sterols was estimated based on the relative percentage of the sterol to ergosterol peak areas in each sample. The mass spectrum of unidentified sterol A and alternative sterol were compared to the authentic standards and spectra profiles from the National Institute of Standards and Technology search database for 4-methyl episterol and lanosterol.

## Illumina whole genome sequencing

Genomic DNA was isolated using a phenol-chloroform extraction as described previously [104]. Libraries were prepared using the Illumina DNA Prep kit and IDT 10bp UDI indices, and sequenced on an Illumina NextSeq 2000, producing 2x151bp reads. Demultiplexing, quality control, and adapter trimming were performed with bcl-convert (https://support.illumina. com/sequencing/sequencing_software/bcl-convert.html)(v3.9.3). Adapter and quality trimming were performed with BBDuk (BBTools v38.94) [105]. Trimmed reads were aligned to the *C. albicans* reference genome (SC5314_version_A21-s02-m09-r08) using BWA-MEM (v0.7.17) with default parameters [106,107]. Aligned reads were sorted, duplicate reads were marked and the resulting BAM file was indexed with Samtools (v1.10) [107]. Quality of trimmed FASTQ and BAM files was assessed for all strains with FastQC (v0.11.7), Qualimap (v2.2.2-dev) and MultiQC (v1.16) [108–110].

## Visualization of whole genome sequencing data

Chromosomal copy number changes were visualized using the Yeast Mapping Analysis Pipeline (YMAP v1.0). Aligned BAM files were uploaded to YMAP and read depth was determined

and plotted as a function of chromosome position using the reference genome *C. albicans* SC5314 (A21-s02-m09-r08). Read depth was corrected for GC-content and chromosome-end bias. WGS data were plotted as the log2 ratio and converted to chromosome copy number (y-axis, 1–4 copies) as a function of chromosome position (x-axis, Chr1-ChrR) using the Yeast Mapping Pipeline (YMAP) [111]. The baseline chromosome copy number (ploidy) was determined by flow cytometry (S1 Table). Haplotypes are indicated by color: gray is heterozygous (AB), magenta is homozygous B, and cyan is homozygous A.

## Variant calling

De novo variant calling and preliminary filtering were performed with Mutect2 and FilterMutectCalls (GATK v4.1.2), both with default parameters as previously described [112]. Variant calling was run separately for 3 groups of strains corresponding to different progenitors. The first group called the Sn152 progenitor as "normal" and Sn152-evolved strains as "tumor". The second group called BWP17 as "normal" and BWP17-evolved strains as "tumor". The third group called P75063 as "normal" and P75063-evolved strains as "tumor". Additional VCF filtering was performed with bcftools (v1.17) [107]. Individual VCF files were subset to remove the progenitor strain, and filtered for calls with a quality status of "PASS". A merged VCF file was created for each progenitor group. Merged VCF files were subset to exclude repeat regions (as marked in the SC5314 A21-s02-m09-r08 GFF) and 5000 bp subtelomeric regions, and additional hard filtering was performed (minimum 5 supporting reads, at least one supporting read in both forward and reverse direction, minimum alternate allele frequency of 0.2 for diploid, single colony cultures). Identical variants found in at least half of all progeny were considered to be present in the progenitor strain and were removed [10]. Variants were annotated with SnpEff (v5.0e, database built from SC5314 version A21-s02-m09-r08, with alternate yeast nuclear codon table) and visually verified in IGV [113,114]. All variants of *ERG251* were compiled into S17 Table.

## Murine model

*C. albicans* strains were serially passaged three times in YPD broth, grown in a shaking incubator at 30˚C for 16-24h at each passage. To prepare *C. albicans* for infection, yeast cells were collected by centrifugation, washed in PBS, and counted using a hemocytometer. Male, 5–6 weeks old ICR mice (Envigo) were infected with 2x10$^5$ *C. albicans* yeast cells via the lateral tail vein. Mice were monitored three times daily for survival for 21 days. Moribund mice were humanely euthanized.

## Supporting information

**S1 Fig. Whole genome sequencing analysis of FLC-evolved and engineered strains. A.** *De novo* point mutations in *ERG251* often occur together with other aneuploidies. Representative whole genome sequencing (WGS) data of the FLC-evolved strains from Table 1: Evolved 3.2, AMS5615, AMS5617, AMS5618, AMS5622, AMS5623, AMS5624, AMS5625, AMS5626 and AMS4130 which acquired point mutations on *ERG251* during FLC evolution. **B.** The engineered *ERG251* mutants remain euploid. WGS data for all *ERG251* mutations engineered into the euploid SC5314 genetic background: the *ERG251* heterozygous point mutants (L113*, W265G, E273*, and *322Y), both heterozygous deletion strains of *ERG251*, two strains with complementation of the heterozygous deletion, and two independent homozygous deletions of *ERG251* (d51 and d70). **A&B** WGS data are plotted as the log2 ratio and converted to chromosome copy number (y-axis, 1–4 copies) as a function of chromosome position (x-axis, Chr1-ChrR). Haplotypes are indicated by color: gray is heterozygous (AB), magenta is

homozygous B, and cyan is homozygous A. The baseline ploidy was determined by propidium iodide staining (S1 Table).
(TIF)

**S2 Fig. Radicicol, an Hsp90 inhibitor, blocks Erg251-mediated tolerance and makes fluconazole fungicidal. A.** Cells from the MIC assay at 48 hr in Fig 1D, with or without radicicol, were plated for viability on YPAD agar plates and imaged after 24 hr incubation. Wild-type SC5314 (*ERG251/ERG251*), and both heterozygous deletion mutants of *ERG251* were tested with a FLC resistant clinical isolate (C17/12-99, S1 Table) as a positive control. **B**. Spot plates growth of *ERG251/ERG251*, *erg251Δ/ERG251*, *ERG251/erg251Δ*, and *erg251Δ/Δ* on YPAD (30˚C), YPAD (37˚C), 20 μg/ml calcofluor white (CFW), 125 μg/ml Congo Red, 1.2M NaCl, 0.03% SDS and 7.5mM $H_2O_2$ agar plates. **A & B**. At least three biological replicates were performed.
(TIF)

**S3 Fig. Heterozygous deletion of *ERG251-A* leads to a transcriptional response in filamentation regulation.** Volcano plots for differentially expressed genes (log2 fold change $\geq 0.5$ or $\leq$ -0.5 and adjusted *p*-value $< 0.1$) in the heterozygous mutants **(A)** *erg251Δ/ERG251* and **(B)** *ERG251/erg251Δ* in YPAD compared to the wild-type *ERG251/ERG251* in YPAD. Both the fold change and *p*-value are indicated. **C&D.** Gene Ontology (GO) terms for genes differentially expressed in (C, S7 Table) *erg251Δ/ERG251* in YPAD and (D, S8 Table) *ERG251/erg251Δ* in YPAD compared to *ERG251/ERG251* in YPAD. **E.** Quantification of the yeast ($<6$μm), pseudohyphae (15–36 μm), and hyphae ($>36$ μm) for both evolved and engineered strains with *ERG251* point mutations on A or B allele together with SC5314 as the control. Evolved strains: *ERG251^{W265G}/ERG251* (AMS5625 and AMS5626) and *ERG251/ERG251^{*322Y}* (evolved 3.2). 150 to 500 cells were counted for each strain, and at least two biological replicates were performed. Values are mean ± SEM calculated from three biological replicates. Statistical significance for filamentation was compared to *ERG251/ERG251* and assessed using two-way ANOVA with uncorrected Fisher's LSD, ****$P < 0.0001$, ns: $P > 0.05$. **F.** Relative expression of *ERG251-A* and *ERG251-B* in the SC5314 background in YPAD. Relative expression was estimated using allelic RNA reads compared to overall reads at the two loci with polymorphisms in the *ERG251* gene (indicated as SNP1 and SNP2 above). Values are mean ± SEM calculated from three biological replicates. **G.** Subcellular localization of Erg251-A-GFP (top) and Erg251-B-GFP (bottom) in yeast and hyphal inducing conditions in SC5314 background. Yeast: scale bar, 5 μm; hyphae: scale bar, 10 μm. ER tracker (red) was applied in Erg251-A-GFP and Erg251-B-GFP yeast cells indicating the ER co-localization.
(TIF)

**S4 Fig. Homozygous deletion of *ERG251* leads to downregulation of ergosterol biosynthesis genes and upregulation of alternate sterol biosynthesis genes. A.** The relative gene expression levels (log2-fold change) for all *ERG* genes in the heterozygous and homozygous mutants *erg251Δ/ERG251*, *ERG251/erg251Δ*, and *erg251Δ/Δ* grown in YPAD or YPAD+1μg/ml FLC conditions, compared to the wildtype *ERG251/ERG251* in the same condition. Two asterisks indicate the expression change is significant (adjusted p-value $< 0.05$) in *erg251Δ/Δ* relative to *ERG251/ERG251* in both YPAD (S2 Table) and YPAD+1μg/ml FLC (S11 Table) conditions. One asterisk indicates *ERG6* expression level change is significant (adjusted p-value $< 0.05$) in *erg251Δ/Δ* relative to *ERG251/ERG251* only in YPAD+1μg/ml FLC condition (S11 Table). **B.** The relative expression level (log2 fold change) of *ERG* genes in the wildtype *ERG251/ERG251*, and mutants *erg251Δ/ERG251*, *ERG251/erg251Δ*, and *erg251Δ/Δ* grown in YPAD+1μg/ml FLC compared to YPAD condition. One asterisk indicates the expression

change is significant (adjusted *p*-value < 0.05) in the *erg251Δ/Δ* in YPAD relative to *erg251Δ/Δ* in YPAD+1μg/ml fluconazole conditions (S15 Table).
(TIF)

**S5 Fig. Putative estimates of unidentified sterols.** Mass spectra of two unidentified sterols: Sterol A (24.6 min) and the Alternate sterol (25.1 min) from Fig 6B, and two related standards Gramisterol (Smolecule, catalog # S626191) and Lanosterol (Smolecule, catalog #S532452).
(TIF)

**S6 Fig. Erg251 is the major methyl sterol oxidase controlling drug susceptibility compared to its paralog Erg25.** (**A**) Multiple sequence alignment for *ERG251-A*, *ERG251-B*, and *ERG25-A/-B* (no SNPs between A and B) from *C. albicans* and *ERG25* from *S. cerevisiae*, with yellow highlighting similarity among all four proteins. Colored blocks on the top indicate the sequence conservation. Asterisks (*) and red boxes indicate the locus of non-synonymous variation between *ERG251-A* and *ERG251-B* in *C. albicans*. **B.** FLC susceptibility determined by liquid microbroth dilution at 24hr MIC (left, μg/ml) and 48hr SMG (right, tolerance) in FLC for three *ERG25* heterozygous deletion mutants (*ERG25/erg25Δ-2*, *-8* and *-10*) in the SC5314 background with SC5314 (*ERG25/ERG25*) as the control. MIC: each dot represents a single replicate and bar represents the average of three biological replicates of a single strain; SMG values are mean ± SEM calculated from three biological replicates of a single strain. **C.** 48hr growth curve analysis of *erg25* heterozygous deletion strain in *erg251Δ/Δ* background (*erg251Δ/Δ*: *ERG25/erg25Δ*) in YPAD (left) and YPAD+1μg/ml FLC (right) with SC5314 (*ERG25/ERG25*) and *erg251Δ/Δ* as the controls. The initial cell densities were $OD_{600}$ of 0.001. MIC and SMG are not measurable for *erg251Δ/Δ* or *erg251Δ/Δ*: *ERG25/erg25Δ* given growth defects in YPAD. **B&C**: Minimum of three biological replicates were performed. **D.** RNA abundance of *ERG251* and *ERG25* in SC5314 (wild-type), and in *erg251Δ/Δ*. RNA reads were normalised to transcript length and total RNA reads. Values are mean ± SEM calculated from three biological replicates. **E.** Predicted model for how FLC and farnesol impact the expression of *ERG* genes. In the wild-type, low concentrations of FLC promotes the expression of most *ERG* genes, including *ERG6*, *ERG251*, *ERG25*, *ERG11* and *ERG27*, leading to the upregulation of ergosterol or/and alternate sterol biosynthesis. However, both low concentrations of FLC and Erg251 pose a negative regulation on Erg12, which may be achieved via farnesol which we predict inhibits *ERG12* [90]. Dashed lines indicate predicted relationships. Figure created in BioRender.com.
(TIF)

**S1 Table. Strains used in this study.**
(XLSX)

**S2 Table. Differentially expressed genes in *erg251Δ/Δ* in YPAD compared to wild-type in YPAD.**
(XLSX)

**S3 Table. GO term analysis for differentially expressed genes in *erg251Δ/Δ* in YPAD compared to wild-type in YPAD.**
(XLSX)

**S4 Table. Differentially expressed GPI genes in *erg251Δ/Δ* in YPAD compared to wild-type in YPAD.**
(XLSX)

**S5 Table. Differentially expressed genes in *erg251Δ/ERG251* in YPAD compared to wild-type in YPAD.**
(XLSX)

**S6 Table. Differentially expressed genes in *ERG251/erg251Δ* in YPAD compared to wild-type in YPAD.**
(XLSX)

**S7 Table. GO term for differentially expressed genes in *erg251Δ/ERG251* in YPAD compared to wild-type in YPAD.**
(XLSX)

**S8 Table. GO term for differentially expressed genes in *ERG251/erg251Δ* in YPAD compared to wild-type in YPAD.**
(XLSX)

**S9 Table. Differentially expressed genes in *erg251Δ/ERG251* in FLC compared to wild-type in FLC.**
(XLSX)

**S10 Table. Differentially expressed genes in *ERG251/erg251Δ* in FLC compared to wild-type in FLC.**
(XLSX)

**S11 Table. Differentially expressed genes in *erg251Δ/Δ* in FLC compared to wild-type in FLC.**
(XLSX)

**S12 Table. Differentially expressed *ERG* genes in wild-type (*ERG251/ERG251)* in FLC compared to wild-type (*ERG251/ERG251)* in YPAD.**
(XLSX)

**S13 Table. Differentially expressed *ERG* genes in *erg251Δ/ERG251* in FLC compared to wild-type *erg251Δ/ERG251* in YPAD.**
(XLSX)

**S14 Table. Differentially expressed *ERG* genes in *ERG251/erg251Δ* in FLC compared to wild-type *ERG251/erg251Δ* in YPAD.**
(XLSX)

**S15 Table. Differentially expressed *ERG* genes in *erg251Δ/Δ* in FLC compared to wild-type *erg251Δ/Δ* in YPAD.**
(XLSX)

**S16 Table. Primers used in this study.**
(XLSX)

**S17 Table. *ERG251* SNPs from all FLC-evolved strains.**
(XLSX)

## Acknowledgments

We are grateful to Berman lab, Cowen lab and Köhler lab for the plasmids used for strain engineering: pMG2120, pLC605 and pJK863. We thank Luke Dragseth, Maicy Vossen and Hanaa Alhosawi for technical assistance with the evolution and sequencing of some of the evolved

strains where *ERG251* mutations were initially identified. We are grateful to Petra Vande Zande for helpful discussions and feedback on the manuscript. We are extremely grateful to the Mitchell lab for sharing and discussing results prior to publication.

## Author Contributions

**Conceptualization:** Xin Zhou, Laura S. Burrack, Anna Selmecki.

**Data curation:** Xin Zhou, Laura S. Burrack.

**Formal analysis:** Xin Zhou, Audrey Hilk, Nivea Pereira De Sa, Laura S. Burrack, Anna Selmecki.

**Funding acquisition:** Maurizio Del Poeta, Laura S. Burrack, Anna Selmecki.

**Investigation:** Xin Zhou, Audrey Hilk, Norma V. Solis, Nivea Pereira De Sa, Bode M. Hogan, Tessa A. Bierbaum, Anna Selmecki.

**Methodology:** Xin Zhou, Norma V. Solis, Laura S. Burrack, Anna Selmecki.

**Project administration:** Anna Selmecki.

**Resources:** Maurizio Del Poeta, Scott G. Filler, Laura S. Burrack, Anna Selmecki.

**Supervision:** Maurizio Del Poeta, Scott G. Filler, Laura S. Burrack, Anna Selmecki.

**Validation:** Xin Zhou, Audrey Hilk.

**Visualization:** Xin Zhou.

**Writing – original draft:** Xin Zhou, Laura S. Burrack, Anna Selmecki.

**Writing – review & editing:** Xin Zhou, Laura S. Burrack, Anna Selmecki.

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
