## [Decision Letter · Decision Letter 0]

18 Mar 2024

Dear Dr. Selmecki,

Thank you very much for submitting your manuscript "Erg251 has complex and pleiotropic effects on azole susceptibility, filamentation, and stress response phenotypes" for consideration at PLOS Pathogens. As with all papers reviewed by the journal, your manuscript was reviewed by members of the editorial board and by several independent reviewers. In light of the reviews (below this email), we would like to invite the resubmission of a significantly-revised version that takes into account the reviewers' comments.

While the reviewers were generally positive about this work and its significance, they have identified a number of issues that need to be addressed. These will require additional experimentation and revision of this manuscript.

The companion submission to your manuscript has also been evaluated on its merits, and we have reached a different decision based on the reviewers' recommendation. Therefore, we cannot guarantee to publish the two submissions side-by-side. 

We cannot make any decision about publication until we have seen the revised manuscript and your response to the reviewers' comments. Your revised manuscript is also likely to be sent to reviewers for further evaluation.

Sincerely,

Chaoyang Xue, Ph.D.

Academic Editor

PLOS Pathogens

Alex Andrianopoulos

Section Editor

PLOS Pathogens

Michael Malim

Editor-in-Chief

PLOS Pathogens

orcid.org/0000-0002-7699-2064

Reviewer's Responses to Questions

**Part I - Summary**

Reviewer #1: In this paper, the authors report that under selective pressure in the presence of fluconazole, C. albicans can become drug-tolerant by the acquisition of heterozygous loss-of-function mutations in ERG251, encoding an enzyme in the ergosterol biosynthesis pathway. In combination with certain chromosome aneuploidies these mutations resulted in azole resistance. Strains lacking one ERG251 allele had pleiotropic phenotypes, but in contrast to a homozygous erg251 deletion mutant (which had much stronger phenotypic defects) retained fitness in the absence of the drug and virulence in a mouse infection model. The authors suggest that alterations in sterol biosynthesis and upregulation of the zinc transporter-encoding gene ZRT2 contribute to the acquired azole tolerance of erg251 mutants.

Main comment

1) The authors studied many phenotypic consequences of the ERG251 mutations, but did not investigate how these affected the lipid composition of cellular membranes, which is the expected primary effect of the altered Erg251 activity. The observed phenotypes of the mutants are most likely secondary consequences of an altered membrane structure. For a mechanistic understanding of drug tolerance and other mutant phenotypes, knowledge of the underlying primary defect of the heterozygous erg251 mutants would be required (phenotypes and transcriptional changes observed in the erg251 null mutants do not explain the behavior of the heterozygous mutants). Interestingly, both reduced and increased ERG251 dosage resulted in azole tolerance (Fig. 1B), but the ERG251-overexpressing strain was not compared with the heterozygous mutants in subsequent experiments and this unexpected result was not further explored and remains unexplained.

Reviewer #2: This manuscript identified a role C. albicans ERG251, a paralog of ERG25, in fluconazole tolerance. An in vitro evolution strategy was used to identify C. albicans strains that are tolerant to fluconazole. Three independent experiments led to the identification of strains with ERG251 heterozygous mutations. The role of ERG251 was confirmed by showing that heterozygous mutants displayed similar fluconazole tolerance. The homozygous erg251/erg251 mutants showed complex phenotypes, including decreased fitness at low initial cell density and increased fitness in the presence of low concentrations of fluconazole. Changes in gene expression detected by RNAseq were used to examine other phenotypes in erg251 mutant strains. These studies showed that the strains were more sensitive to SDS, weakly resistant to H2O2, and displayed a weak hyphal defect. Many ergosterol biosynthesis genes were down-regulated in the erg251D/D strain but the azole tolerance of the heterozygous ERG251 mutant did not appear to be due to changes in ergosterol biosynthesis gene expression. The ZRT2 zinc transporter was upregulated in erg251 mutants, and control studies suggest this may contribute the fluconazole phenotypes. The heterozygous mutants were virulent in a mouse disseminated infection, but the homozygous mutant showed a defect in virulenc3e. Overall, the strength is that these studies discovered a role for ERG251 in promoting fluconazole tolerance in vitro. Weaknesses include the lack of in-depth studies to characterize phenotypes and to better define the role of ERG251 in tolerance.

Reviewer #3: I really liked all the beginning parts of this paper where the authors told me why I should be interested in ERG251 as a hotspot for studying antibiotic resistance. It was novel and very convincing. Then we come to the experiments characterizing the single and double knockout mutants of erg251. The results here are interesting, bordering on fascinating, but I found it difficult to consider them in the proper context. Fig 6A did not help in this regard. I would like to see a clear presentation of where you think Erg25p and 251p fit into biosynthesis. In reading Lu et al (2023), they had Erg25p in the main sterol biosynthesis pathway and Erg251p in the alternative pathway, but you seem to be avoiding such a clear distinction. I think this portion of the text would be more easily understood if accompanied by a diagram presenting your best thoughts on how the 251p fits into biosynthesis.

Similarly, Lu et al stated that the erg251 double null mutant failed to grow in the presence of fluconazole (their Fig 4A-C) while you have the double 251p mutant presenting a fitness advantage at low fluconazole concentrations and a fitness cost at high fluconazole concentrations. Is this strictly a difference in concentrations chosen, or does it present something more fundamental? To me, such differences suggest multiple targets for the azole inhibitors. Both the Erg25p and 251p are C-4 sterol methyl oxidases. Candida albicans and Aspergillus fumigatus have two such enzymes while S. cerevisiae and the ergosterol containing algae Chlorella (Voshall et al 2021) and Chlamydomonas (Brumfield et al 2017) have only one. There are two methyl groups on the C4 position of lanosterol which need to be removed. Perhaps there is a division of labor. Voshall et al examined the sterols which were produced by Chlorella following inhibition by either ketoconazole or clotrimazole (fluconazole was not inhibitory). The relevant feature for your paper was that identifying the unusual or overflow sterols produced following these antibiotic treatments strongly suggested that four different sterol precursors were available as substrates for the single C4-sterol methyl oxidase. Could Erg25p and 251p exhibit different substrate preferences? Please note that I am not suggesting detailed sterol compositions for each of your mutants following antibiotic treatment. Those experiments might be desirable in the long term but highly impractical in the short term. I'm merely seeking clarification on your thinking on the respective roles of Erg25p and Erg251p.

Reviewer #4: The manuscript by Zhou et al describes in detail the discovery that growth of C. albicans strains in fluconazole can lead to the development of single allele ERG251 mutations that increase fluconazole tolerance. When these heterozygous ERG251 mutations were combined with spontaneous aneuploidies in chromosomes 3 or 6, they also saw full resistance to fluconazole. In addition, they characterized the impacts of heterozygous ERG251 mutations on either or both alleles for their impacts on growth rate, global gene transcription, hyphal growth, cell wall sensitivities, and virulence in mice. Their findings implicated ERG251 in impacting recovery from lag phase at low density growth through affecting farnesol. They found that one allele of ERG251 was more effective than the other for several phenotypes like hyphal growth and transcription of other ERG genes. They also found that the ZRT2 zinc transporter’s transcription was impacted by ERG251 and could affect azole sensitivity of wild-type when overexpressed. Finally, the erg251∆∆ mutant was diminished in virulence, but the heterozygotes were not.

Overall, I found this manuscript to have a wide breadth of coverage of many different phenotypes, but several at a mostly superficial level. There was not a clear model for how Erg251 affects these many different phenotypes such as transcription or hyphal growth.

**Part II – Major Issues: Key Experiments Required for Acceptance**

Reviewer #1: Other comments

2) To test whether the slight upregulation of ZRT2 contributed to the fluconazole tolerance of erg251 mutants, the authors overexpressed ZRT2 in a wild-type strain (overexpression levels should be given and compared to those in erg251 mutants). To more directly address the question and support the conclusion stated in the paragraph title (lines 560-561), one could test if ZRT2 deletion (possibly only one allele if this reduces expression levels to those in the wild type) in erg251 mutants reverts tolerance.

3) Is the allele-specific effect of ERG251 on filamentation in the SC5314 background also seen in the evolved strains (SN152 and BWP17 are derivatives of SC5314), i.e. does Evolved 3.2 behave like the heterozygous mutant in which allele B was deleted, and do AMS5617/5618, AMS5622/5623/5624, and AMS5625/5626 behave like the heterozygous mutant lacking allele A? Furthermore, did introduction of the loss-of-function mutations into alleles A or B of strain SC5314 (lines 194-198 and Fig. 1A) have the same allele-specific effect on filamentation?

Reviewer #2: 1. The legends to Fig. 1 and 2 state that each bar represents the average of three technical replicates. Were these studies repeated in independent experiments? No error bars are shown.

2. Fig. 3A. Is this a representative curve? Average of three assays? What were the lag phases for the three assays? Do erg251 deletion mutants grow better in conditioned medium?

3. Figure 4 is not supported by strong data.

(i) A slight increase in sensitivity to SDS could be due to many reasons, such as altered membrane lipids. It is not specifically indicative of altered cell wall.

(ii) Increased resistance to oxidative stress is not well supported. There appears to be a very weak effect in a spot assay. This should be quantified. The magnitude of the resistance does not seem very significant. Was it reproducible?

4. Figure 5. There appears to be a slight defect in hyphal growth for ERG251-A deletion and a little stronger defect for erg251D/D mutant. However, it is not clear that this is significant. Only limited characterization was presented. Also, it was not clear that there was a defect in vivo. No analysis of hyphal growth in vivo was presented.

5. Figure 6 is very descriptive. It shows that ERG gene expression is altered, but it is not clear if this is significant.

6. Figure 7. Lines 591-596. The conclusion that altered ZRT2 expression contributes to fluconazole tolerance of erg251/ERG251 mutants is not supported by strong data. The TET-promoter strain showed increased tolerance to fluconazole, but it was not clear what was the level of ZRT2 expression in the Tet-ZRT2 strain. Also, it was not clear how the authors concluded that the ZRT2 effect is distinct from distinct from the ATP-dependent drug efflux pumps such as CDR1.

7. Figure 8. Mouse virulence. It would have been interesting to see if there was a difference in ability of fluconazole to prevent lethal infection. Perhaps a mixed infection to see if the heterozygotes have a better ability to survive? Some type of experiment like that would have strengthened the conclusions.

8. The Discussion section contained a lot of speculation about minor effects that could be shortened.

Lines 755 -765. Not proven ZRT2 was overexpressed, or expressed at the level seen in the erg251 mutant.

Reviewer #3: None

Reviewer #4: 1. For example, the overexpression study with ZRT2 implicates it in being an effector of ERG251 fluconazole sensitivity, but additional work would be needed to draw this conclusion definitively. Overexpression of ZRT2 clearly impacts this phenotype, but it could be a parallel pathway rather than a direct linear effect related to ERG251.

2. The localization of ERG251-GFP needs to be better controlled. Were these alleles able to complement an erg251∆∆ mutant? Colocalization with a known ER marker needs to be done make the conclusion of ER localization

3. The conclusion drawn in lines 222-225: “We found that ERG251 -drivenazole tolerance was independent of drug efflux pumps as indicated by no increase in the rate of efflux of R6G for ERG251 heterozygous deletion mutants compared to ERG251/ERG251 (SC5314) during the exposure to FLC (Fig 1C).” is not consistent with the data shown in Fig 1C where it appears that mutants are progressively less able to excrete the drug than the wild-type.

4. In Figure 4, SDS is a better measure of membrane stress, but not cell wall stress. Specific stressors of the cell wall such as echinocandins, calcofluor white, and Congo Red should be tested.

**Part III – Minor Issues: Editorial and Data Presentation Modifications**

Reviewer #1: 4) The mutation *322Y (line 174 and Table 1) is incorrectly describe

---

## [Decision Letter · Decision Letter 1]

3 Jul 2024

Dear Dr. Selmecki,

We are pleased to inform you that your manuscript 'Erg251 has complex and pleiotropic effects on sterol composition, azole susceptibility, filamentation, and stress response phenotypes' has been provisionally accepted for publication in PLOS Pathogens.

Best regards,

Chaoyang Xue, Ph.D.

Academic Editor

PLOS Pathogens

Alex Andrianopoulos

Section Editor

PLOS Pathogens

Michael Malim

Editor-in-Chief

PLOS Pathogens

orcid.org/0000-0002-7699-2064

Reviewer Comments (if any, and for reference):

Reviewer's Responses to Questions

**Part I - Summary**

Reviewer #1: The authors have thoroughly addressed my previous criticisms, and I congratulate them for an excellent paper.

Reviewer #3: This revised manuscript has been significantly improved. I am pleased that they did the sterol assays to help interpret their mutant analyses.

Reviewer #4: The manuscript by Zhou et al describes in detail the discovery that growth of C. albicans strains in fluconazole can lead to the development of single allele ERG251 mutations that increase fluconazole tolerance. When these heterozygous ERG251 mutations were combined with spontaneous aneuploidies in chromosomes 3 or 6, they also saw full resistance to fluconazole. In addition, they characterized the impacts of heterozygous ERG251 mutations on either or both alleles for their impacts on growth rate, global gene transcription, hyphal growth, cell wall sensitivities, and virulence in mice. Their findings implicated ERG251 in impacting recovery from lag phase at low density growth through affecting farnesol. They found that one allele of ERG251 was more effective than the other for several phenotypes like hyphal growth and transcription of other ERG genes. They also found that the ZRT2 zinc transporter’s transcription was impacted by ERG251 and could affect azole sensitivity of wild-type when overexpressed. Finally, the erg251∆∆ mutant was diminished in virulence, but the heterozygotes were not. The concerns I had previously have been addressed.

**Part II – Major Issues: Key Experiments Required for Acceptance**

Reviewer #1: (No Response)

Reviewer #3: None

Reviewer #4: (No Response)

**Part III – Minor Issues: Editorial and Data Presentation Modifications**

Reviewer #1: Considering that azole tolerance is a frequent cause of treatment failure in the clinic (as the authors point out), a critical reflection on why erg251 mutations have not been found in azole-tolerant clinical isolates so far would have been desirable. Or have they? I assume the authors have searched for such mutations in the many C. albicans genome sequences that are available nowadays.

Reviewer #3: None

Reviewer #4: (No Response)

PLOS authors have the option to publish the peer review history of their article (what does this mean?). If published, this will include your full peer review and any attached files.

Reviewer #1: No

Reviewer #3: No

Reviewer #4: No

---

## [Editor Report · Acceptance letter]

25 Jul 2024

Dear Dr. Selmecki,

We are delighted to inform you that your manuscript, "Erg251 has complex and pleiotropic effects on sterol composition, azole susceptibility, filamentation, and stress response phenotypes," has been formally accepted for publication in PLOS Pathogens.

Best regards,

Michael Malim

Editor-in-Chief

PLOS Pathogens

orcid.org/0000-0002-7699-2064